



# Precipitation Susceptibility in Marine Stratocumulus and Shallow Cumulus from Airborne Measurements

Eunsil Jung[1], Bruce A. Albrecht[1], Armin Sorooshian[2,3], Paquita Zuidema[1], Haflidi H. Jonsson[4]

[1]Department of Atmospheric Sciences, University of Miami, Miami, FL, 33149, United States
[2]Department of Chemical and Environmental Engineering, University of Arizona, Tucson, AZ, 85721,USA
[3]Department of Hydrology and Atmospheric Sciences, University of Arizona, Tucson, AZ, 85721, USA
[4]Naval Postgraduate School, Monterey, CA, 93943, USA

*Correspondence to:* Eunsil Jung (eunsil.jung@gmail.com)

**Abstract.** Precipitation tends to decrease as aerosol concentration increases in warm marine boundary layer clouds at fixed liquid water path (LWP). The quantitative nature of this relationship is captured using the precipitation susceptibility ($S_o$) metric. Previously published works disagree on the qualitative behavior of $S_o$ in marine low clouds: $S_o$ decreases monotonically with increasing LWP or cloud depth (H) in stratocumulus clouds (Sc), while it increases and then decreases in shallow cumulus clouds (Cu). This study uses airborne measurements from four field campaigns on Cu and Sc with similar instrument packages and flight maneuvers to examine if and why $S_o$ behavior varies as a function of cloud type. The findings show that $S_o$ increases with H and then decreases in both Sc and Cu. Possible reasons for why these results differ from those in previous studies of Sc are discussed.

## 1 Introduction

Cloud-aerosol interactions are considered to be one of the most important forcing mechanisms in the climate system (IPCC, 2013). It is believed that aerosols suppress precipitation in warm boundary layer clouds. However, there is considerable disagreement on the magnitude and even in the sign of aerosol perturbations on cloud feedback such as cloud fraction and lifetime (Stevens and Feingold, 2009). Furthermore, aerosol effects on cloud and precipitation are not readily separable from the effects of meteorology on cloud and precipitation. The precipitation susceptibility metric, $S_o$, quantifies how aerosol perturbations alter the magnitude of precipitation rate (R) while minimizing the effects of macrophysical factors (i.e., meteorology) (Feingold and Siebert, 2009). It is defined as

$$S_o = -\frac{d \ln R}{d \ln N_d} , \qquad (1)$$

and is evaluated with fixed cloud macrophysical properties, such as cloud thickness (H) or liquid water path (LWP). In Eq. (1), aerosol effects are embedded in the cloud droplet number concentration ($N_d$) variable since aerosols serve as cloud





condensation nuclei (e.g., as aerosol concentration increases, $N_d$ increases). The minus sign is used in Eq. (1) to achieve a positive value of $S_o$ due to the expectation that increasing aerosols reduce precipitation (all else fixed). In the original work of $S_o$ (Feingold and Siebert, 2009), cloud-base R and $N_d$ are used. Since then, slightly different definitions of $S_o$ have been adapted to present $S_o$, depending on the dataset used. For example, Sorooshian et al. (2009) used an aerosol proxy (e.g.,

Aerosol Optical Depth and Aerosol Index) instead of $N_d$ for satellite data analysis. Terai et al. (2012, 2015) introduced precipitation susceptibility using the sum of the susceptibilities of drizzle intensity ($S_I$) and drizzle fraction ($S_f$), $S_R=S_I+S_f$, where $S_f$ is equivalent to $S_{pop}$ (probability of precipitation) and $S_I$ is analogous to $S_o$ but for precipitation rate of those clouds that are precipitating. The POP is defined as the ratio of the number of precipitating events over the total number of cloudy events. The $S_{pop}$ is used in some studies of precipitation susceptibility (e.g., Wang et al., 2012; Mann et al., 2014; Terai et al.,

2015). In addition to the different definitions of precipitation susceptibility, various forms of R and $N_d$ (e.g., cloud-base, vertically integrated, or ground-based values, so on) with different data thresholds have been used for calculation of the precipitation susceptibility depending on the data available. In this study, precipitation susceptibility indicates $S_o$ unless otherwise stated.

In global climate models (GCMs), these aerosol effects on precipitation are represented by the collision-coalescence

process (see Sect. 3.2), which is parameterized by a power-law relationship among R, LWP, and $N_d$ as,

$$R = LWP^{\alpha} N_d^{-\beta} .$$

(2)

Climate models currently assume a fixed value of autoconversion parameter ($\beta$ in Eq. 2), ranging between approximately 0 and 2 (e.g., Rasch and Kristjansson, 1998; Khairoutdinov and Kogan, 2000; Jones et al., 2001; Rotstayn and Liu, 2005; Takemura et al., 2005). $S_o$ in Eq. (1) is equivalent to the exponent $\beta$ in Eq. (2) at fixed LWP. For field studies of precipitating

Sc clouds, $\beta$ has been reported in the range of 0.8 to 1.75 at fixed LWP (e.g., Pawlowska and Brenguier, 2003; Comstock et al., 2004; vanZanten and Stevens, 2005; Lu et al., 2009). However, these single power law fits do not show the changes in $S_o$ with LWP or H, which is important since previous works have revealed that clouds within specific ranges of LWP (or H) are more susceptible to aerosol perturbations in terms of changes in precipitation.

The qualitative behavior of $S_o$ has been studied for low clouds using models, remote sensing data, and in situ

measurements. For model studies of warm cumulus clouds (e.g., the adiabatic parcel model of Feingold and Siebert, 2009), $S_o$ varies from 0.5 to 1.1 with increasing LWP, and exhibits three regimes. At low LWP, $S_o$ is insensitive to aerosol perturbation where clouds do not precipitate. At intermediate LWP, suppression of collision-coalescence by the increased aerosols is most effective. We will refer to this regime as the ascending branch of $S_o$ following Feingold et al. (2013). At high LWP, $S_o$ decreases with increasing LWP (hereafter the descending branch of $S_o$). This LWP-dependent pattern of $S_o$ is

supported by satellite observations (Sorooshian et al., 2009; 2010) and large-eddy simulations (LES) (Jiang et al., 2010) for warm trade cumulus clouds. In contrast, Terai et al. (2012) showed that $S_R$ monotonically decreased with increasing LWP and H in Sc clouds by using in situ measurements during VOCALS field study, while their $S_I$ value, which is similar to $S_o$ in



aforementioned studies, did not reveal any significant change with H and maintained a value of ~0.6. These inconsistent results have raised questions of cloud type-dependent behaviors of $S_o$.

To begin to unravel why differences in the various studies exist, Feingold et al. (2013) showed in modeling studies that the time available for collision-coalescence ($t_c$) is critical for determining the LWP-dependent behavior of $S_o$, and may be at least partly responsible for some of the differences. Gettelman et al. (2013) also showed how the microphysical process rates impact the $S_o$ in a global climate model (CAM5 GCCM). They showed that the behavior of $S_o$ with LWP differs in the GCCM and in the steady-state model (Wood et al., 2009); The values of $S_o$ were constant or decreased with LWP in a steady state model (consistent with Terai et al., 2012; Mann et al., 2014), whereas LWP-dependent $S_o$ behavior was found in the GCCM (consistent with Feingold and Siebert, 2009; Sorooshian et al., 2009, 2010; Jiang et al., 2010; Feingold et al., 2013). In their study, altered microphysical process rates significantly changed the values of $S_o$, but the qualitative behavior of $S_o$ with LWP remained unchanged (i.e., $S_o$ increases with LWP and peaks at intermediate LWP then decreases with LWP). More recently, Mann et al. (2014) analyzed 28 days of data from the Azores ARM Mobile Facility where the prevalent type of clouds are cumulus (20 %), cumulus under stratocumulus (10-30 %) and single-layer stratocumulus (10 %), and showed that $S_{pop}$ slightly decreased with LWP. Terai et al. (2015) estimated precipitation susceptibility ($S_I+S_{pop}$) in low-level marine stratiform clouds, which included stratus and stratocumulus clouds, using satellite data. The values of $S_o$ in their study showed similar behaviors of $S_o$ in Mann et al. (2014) in general.

This study is motivated by the inconsistent behavior of $S_o$ in previous studies for warm boundary layer clouds. The focus of this paper is to examine and compare the qualitative behavior of $S_o$ in Cu and Sc using airborne measurements across four field campaigns. Two of them were focused on Sc clouds (VOCALS-Rex and the Eastern Pacific Emitted Aerosol Cloud Experiment, Sect. 2.2) and two of them targeted Cu clouds (Barbados and Key West Aerosol Cloud Experiments, Sect. 2.3). The strength of these four field campaign's airborne measurements is that the same research aircraft was deployed in the campaigns with a similar flight strategy and instrument packages, and provided some uniformity that facilitates comparative analysis. Data and methods are discussed in Section 2, followed by results and discussion in Sections 3 and 4, respectively. The findings are summarized in Section 5. Acronyms used in this study are listed in Table A1 in the Appendix.

## 2 Data and methods

### 2.1 TO aircraft

The Center for Interdisciplinary Remotely Piloted Aircraft Studies (CIRPAS) Twin Otter (TO) research aircraft served as the principal platform from which observations for these four experiments were made. During these four deployments, the TO had similar instrument packages, and made same flight maneuvers in the vicinity of clouds, including vertical soundings and



level-leg flights below, inside, and above the clouds. Each flight had a duration of ~3-4 hours. TO included the following three *in-situ* probes for characterizing aerosol, cloud, and precipitation size distributions: Passive Cavity Aerosol Spectrometer Probe (PCASP), Cloud Aerosol Spectrometer (CAS) and Cloud Imaging Probe (CIP) that resolve particles in diameter ranges from 0.1–2.5 µm, 0.6-60 µm and 25-1550 µm, respectively. A 95-GHz Doppler radar was mounted on top

of the aircraft and detected cloud and precipitation structures above the aircraft. Detailed information of the instruments on the TO and flight strategies is provided elsewhere (Zheng et al., 2011; Jung, 2012).

$S_o$ is calculated from Eq. (1) with $H$ as the macrophysical factor, which was held fixed. H was estimated as the height difference between cloud tops and bases. Cloud tops were determined by the cloud radar with the resolutions of 3 Hz in time and 24 m (5 m) in height for Cu (Sc) while the TO was flying near the cloud base (cloud-base level-leg flight). Cloud

bases of Cu were determined from Lifting Condensation Level (LCL), which were calculated from the average thermodynamic properties of the sub-cloud layer for a given day. Since the cloud-base level-leg flight was designed to fly as close as possible to the cloud-base, and further, LCL varied little for Cu (e.g., Nuijens et al., 2014), $S_o$ was also estimated by using the heights from the cloud-base level-leg flights as the cloud bases, and the results were robust (e.g., see Fig. 4.9 in Jung (2012) for the Barbados Aerosol Cloud Experiment).

In stratocumulus clouds, cloud tops are well defined due to the strong capping inversion (see Zheng et al., 2011); however, cloud bases vary more than tops (e.g., Fig. 2 of Bretherton et al., 2010). As a result, the way that the cloud-base is determined may affect $S_o$ since the changes in cloud base alternatively can change the cloud thickness. Therefore, we estimate $S_o$ with three differently defined cloud bases. In this study, cloud bases of Sc are determined first from LCL calculated from the average thermodynamic properties of the sub-cloud layer (shown as cb-lcl in Fig. 4). Second and third

(cb-lcl and cb-mean), cloud bases are determined from the lowest heights where the vertical gradients of liquid water contents (LWC) are the greatest from the LWC profiles, where the LWC profiles are obtained i) when the aircraft enters the cloud decks to conduct level legs (cb-local), and ii) from the nearest one or two soundings to the cloud-base level-leg flights. The average height of these two lowest heights (cb-mean) is used in this study, along with cb-lcl and cb-local (Fig. 4 later). In general, the heights approximately corresponded to the lowest heights that the liquid water contents (LWC) exceeded 0.01

g m$^{-3}$. $S_o$ was also estimated by using the heights from the cloud-base level-leg flights as the cloud bases, and the qualitative behavior of $S_o$ was preserved (not shown).

$N_d$ and R were calculated from the drop size distribution (DSD), which is obtained from CAS (forward scattering) and CIP probes during the cloud-base level-leg flights, respectively. CAS and CIP probes acquire data every 1 second, and cloud radar receives data every 3 Hz. Therefore, $N_d$, R and H in Eq. (1) were calculated in 1 s resolution (except for

VOCALS-Rex, see Sect. 2.4). R is defined as

$$R = \frac{\pi}{6} \int_{25\mu m}^{1550\mu m} N(D) D^3 u(D) dD , \qquad (3)$$





where $u(D)$ (m s$^{-1}$) is the terminal fall velocity following Gunn and Kinzer (1949) and $N(D)$ (m$^{-3}$ mm$^{-1}$) is the number of drops in a unit volume and given diameter interval dD centered at D (mm). Radar reflectivity Z is calculated from drop size distributions that were obtained from CAS and CIP probes. CAS DSD and CIP DSD were combined to include cloud droplets, rain embryo and drizzle drops, Z=10log(z), where

$$z = \int_{0.6\mu m}^{1550\mu m} N(D)D^6 dD , \qquad (4)$$

in units of mm$^6$m$^{-3}$.

## 2.2 Stratocumulus cloud field campaigns: VOCALS-Rex and E-PEACE

From October to November 2008, the VAMOS Ocean-Cloud-Atmosphere-Land Study-Regional Experiment (VOCALS-REx) took place over the Southeast Pacific (69°W-86°W, 12°S-31°S), an area extending from the near coastal region of

northern Chile and southern Peru to the remote ocean (Zheng et al., 2011; Wood et al., 2011; also see Fig. 1). Three aircrafts were deployed during VOCALS from 14 October to 15 November (NSF/NCAR C-130, DOE G-1, CIRPAS TO). As one of the observational platforms, the TO sampled marine stratocumulus decks in the near coastal region of the VOCALS domain of 20 °S 72°W (Fig. 1). Readers should note that the data Terai et al. (2012) used for $S_o$ calculations, with which we will compare with later, were also obtained from VOCALS. However, their results were based on NSF/NCAR C-130 flights that

sampled cloud decks away from the coastal area (Fig. 1). In the Southeast Pacific Sc decks, the intensity and frequency of drizzle tends to increase westward from the coast (Bretherton et al., 2010). Wood et al. (2011) provide a comprehensive description of VOCALS and Zheng et al. (2011) for the TO aircraft data during the VOCALS. Thirteen out of eighteen VOCALS flights were analyzed to consider typical Sc decks exclusively, by excluding data obtained from flights with decoupled boundary layers, abnormally higher cloud bases, and moist layers above cloud tops (Table 1).

20        From July to August 2011, the Eastern Pacific Emitted Aerosol Cloud Experiment (E-PEACE) took place off the coast of Monterey, California to better understand the response of marine stratocumulus to aerosol perturbations (Russell et al., 2013). E-PEACE combined controlled releases of i) smoke from the deck of the research vessel *Point Sur*, ii) salt aerosol from the research aircraft (TO), and iii) exhaust from container ships transiting across the study area (see Fig. 2 from Russell et al., 2013). During nine out of thirty E-PEACE flights, salt powder was directly introduced into the cloud decks to examine

the effects of Giant cloud condensation nuclei (GCCN) on the initiation of warm precipitation (Jung et al., 2015). After excluding the seeding cases and the non-typical Sc decks, 13 flights remained from which we analyzed data (Table 1). Detailed information about E-PEACE and TO data can be found in Russell et al. (2013).





### 2.3 Marine cumulus cloud field campaigns: BACEX and KWACEX

Shallow marine cumulus clouds are by far the most frequently observed cloud type over the Earth's oceans, yet poorly understood, and have not been investigated as extensively as the other major oceanic warm cloud, Sc. The marine environments in the Caribbean Sea and the Atlantic Ocean provide an excellent area to sample shallow marine cumulus

clouds with a high propensity to precipitate. In addition, African dust is transported from Africa to Miami periodically via North-Atlantic, and affects the clouds over this area (Africa-Barbados-Key West-Miami), providing an excellent opportunity to observe aerosol-cloud-precipitation interactions. To better understand such interactions in these trade cumuli regimes, the Barbados Aerosol Cloud Experiment (BACEX) was carried out off the Caribbean island of Barbados during mid March and mid April 2010 (Jung et al., 2013), and the Key West Aerosol Cloud Experiment (KWACEX) during May 2012 near Key

West (Fig. 1). For the BACEX, we analyzed 12 flights (Table 1). Readers are referred to Jung et al. (2016) for detailed information about the cloud and aerosol properties during the BACEX. The marine atmosphere during KWACEX was dry overall and six out of 21 flights involved sampling in shallow marine cumulus clouds, among which four had sufficient data for analysis (Table 1).

### 2.4 $S_o$ calculation details

The distribution of $N_d$ and R, with the corresponding H, is shown in Fig. 2 for each field campaign as scatter diagrams of $N_d$ and R. All data shown in Fig. 2 were obtained during the cloud-base level-leg flights. Figure 2 essentially shows that as $N_d$ increases, R decreases. The marine environments of Southeast Pacific (SEP) Sc decks (VOCALS, Fig. 2a) were overall drier and more polluted than those in Northeast Pacific (NEP) Sc decks (E-PEACE, Fig. 2c); R=0.03 mm day$^{-1}$ (median) and $N_d$ =253 cm$^{-3}$ in VOCALS, but R=1.04 mm day$^{-1}$ and $N_d$=133 cm$^{-3}$ in E-PEACE. In a few cases, high $N_d$ is observed during E-

PEACE (e.g., $N_d > 400$ cm$^{-3}$ in Fig. 2c), and they are likely associated with the emitted aerosols from the ship exhaust and smoke (Russell et al., 2013; Wang et al., 2014; Sorooshian et al., 2015). The marine environments of the Caribbean Seas (Fig. 2b and Fig. 2d) showed wide variations of R. In Fig. 2, Barbados shows the most pristine environments ($N_d < 350$ cm$^{-3}$, $N_d = 61$ cm$^{-3}$ on average), reflecting the isolated location of the island in the North Atlantic even though the experiment period included the most intense dust events during the year of 2010 (Jung et al., 2013). On the other hand, marine

environments near Key West show polluted conditions (Fig. 2d, $N_d = 206$ cm$^{-3}$ on average).

     $S_o$ is found to be about 0.62 for E-PEACE, if it is calculated by using all the individual data points shown in Fig. 2 where H ranges from ~100 m to 500 m. However, $S_o$ is about 0.42 if one rainy day (shown as double circles in Fig. 6 later) is excluded from the analysis, suggesting the possible artifacts of wet scavenging effects (see Sect. 4) or the influence of macrophysical properties other than H. E-PEACE $S_o$ agrees with values estimated in previous campaigns in the same study

region for H ~200-600 m; $S_o$ ~0.46-0.48 using H and $S_o$ ~0.60-0.63 using LWP (Lu et al., 2009). $S_o$ during VOCALS is about 1.07 for H ~ 150-700 m. Overall, $S_o$ values in this study are within the range of $S_o$ from the previous field studies of



precipitating stratocumulus clouds ($S_o$ ~0.8 to 1.75 for a fixed LWP in the studies of Pawlowska and Brenguier, 2003; Comstock et al., 2004; vanZanten and Stevens, 2005). The value of $S_o$ for BACEX and KWACEX is about 0.89 and 0.77, respectively. The values of $S_o$ for marine cumulus clouds from the field studies have not been reported yet in previous studies.

The single power law fits by using all the data points for a given field campaign give the general sense of $S_o$ values, but do not show the qualitative behavior of $S_o$ with H, which reveals which thickness is most susceptible to aerosol perturbations. In this study, to examine the qualitative behavior of $S_o$ with H, for each campaign, cloud data (1 s resolution of $N_d$, R, and H) were assigned to specific H intervals first. Then, the linear regression line was obtained for a given H interval, on the log ($N_d$) and -log (R) diagram. An example of $S_o$ is shown in Fig. 3 for the E-PEACE. Every single cloud data point

(i.e., $N_d$ and R) for H between 160 m and 190 m is shown in Fig. 3(a). The slope (i.e., linear fit) in Fig. 3a corresponds to the $S_o$ value of 0.24. The value of $S_o$ (0.24) is then plotted in the corresponding H on the H- $S_o$ diagram (e.g., Fig. 4 at the H of 174 m, which corresponds to the average H of the interval). The same procedure is repeated for all H intervals to obtain the complete pattern of $S_o$ with H. We tested and applied a few criteria in the $S_o$ calculations, such as minimum R thresholds, and the total number of cloud data points and spans of $N_d$ for a given H interval. We noted that $S_o$ tended to be unphysically high

in the case of small $N_d$ variations (i.e., short spans of $N_d$) (not shown). Based on these sensitivity tests, it was decided that $S_o$ would be calculated exclusively if dlog($N_d$) spans at least 2.2 and the number of data points exceeds six for a given H interval. For example, in Fig. 3a, dlog($N_d$) spans about 3.5 and the number of data points exceeds 444. Slightly different and broad criteria were applied for Cu mainly due to the larger number of data points sampled in Sc. However, the qualitative behavior of $S_o$ was robust as long as the variation of $N_d$ is large enough, regardless of the other criteria, although the details were different (e.g., Figs. 1-2 in the Appendix). Most of the slopes are statistically significant at the 99 % confidence level

(e.g., filled symbols in Fig. 4). The number of data points used to calculate the value of $S_o$, and the (statistically significant) level of confidence for the fitted line, for a given H interval, is summarized in Table A2.

Additionally, $S_o$ is calculated by resampling the cloud data every nth interval to examine how sample size affects $S_o$ values. For example, in Fig. 3(b), $S_o$ is calculated from the subsets of data that are sampled every 4[th] interval (every 4

seconds, equivalently). That being said, cloud data every ~200-240 m apart from one to the next were used to calculate $S_o$ (the aircraft speed during E-PEACE was about 50-60 m s[-1], and thus, every 4th interval corresponds to ~200-240 m distance in general). The ranges of $S_o$ are shown as vertical bars in Fig. 4 where $S_o$ was calculated from the subsets of data with a sample size ranging from 1/n to 1/10 of the initial sample size (i.e., from n=1 to n=10). $S_o$ values that are calculated with a sample size of n=4 and n=7 are shown in Fig. 3(b-c) as an example. In Fig. 3(b-c), $S_o$ tends to be overestimated as the sample

size decreases (i.e., the spatial and temporal resolution decreases).

$S_o$ during VOCALS is calculated in slightly different ways from other experiments since the cloud radar failed during the VOCALS. First, H is estimated from the vertical structure of LWC for each day (daily mean H). Once H is





determined for each flight, it is assigned to a certain H group with similar H values. For example, H of 9 Nov. (164±18 m) and 10 Nov. (194±21 m) are similar and thus assigned to the same H group (group 1 in Table 1). It should be noted that the daily H differs one from another days, but the differences are not substantial as the each H represents an average value of H on a given day. Accordingly, VOCALS H is classified into only four groups with a similar daily H. Once $N_d$ and R are

assigned to the corresponding H, $S_o$ then is estimated by using all the data points that are assigned to the same H group.

LWP has commonly used as the macrophysical factor when quantifying Eq. (1). However, in this study, we use H as a macrophysical factor since we aim to compare $S_o$ for both Sc and Cu. Further, H is considered to be in good agreement with LWP (LWP ~ $H^2$). The adiabatic assumption, which may be valid in Sc, is not valid in Cu (Rauber et al., 2007; Jung et al., 2016) to calculate LWP. Nevertheless, if we calculate LWP by integrating LWC with height, we would obtain one LWP

profile that could be used for the entire cloud layer on a given day, which is inferior to H that was estimated from the cloud radar with higher temporal resolution (3 Hz versus daily). Moreover, during the campaign, the TO did not carry an instrument that measures LWP directly such as G-band Vapor Radiometer (e.g., Zuidema et al., 2012). Consequently, the direct comparison with previous results of $S_o$ with LWP (e.g., quantitative) is not possible, except for the qualitative behavior of $S_o$ with H, which is the subject of this study. It should be noted that the LWC decreases as the drizzle rates increases (e.g.,

see Fig. 8d of Jung et al., 2015). Furthermore, clouds that are precipitating (higher R) may have a LWP that is lower than the adiabatic value, and a cloud with a small R may have a LWP close to the adiabatic value. It should be also noted that the ranges of H (and possibly LWP) differ substantially between Cu and Sc. For example, H of Cu in this study can be as high as 1700 m, whereas H of Sc is generally less than 500 m (e.g., Fig. 4). Additionally, H for clouds that begin to precipitate may differ in Sc and Cu. In a case of Cu, cloud tends to precipitate once H is deeper than 500-600 m (see Jung et al., 2016,

however, note that H in their study is determined from the height difference between the cloud base that is the height of cloud-base level-leg flight and the cloud top that is measured from the cloud radar, and thus, H in this study is likely slightly higher than H in their study). Besides, the LWP for clouds that precipitate would be sub-adiabatic and would have a smaller value of LWP than the LWP for clouds that are not precipitating. Consequently, $S_o$ that is calculated from the cloud fields where more than one cloud type exists (e.g., Mann et al., 2014; Terai et al., 2015) may be complicated since LWP is shifted

to smaller values for (heavily) precipitating clouds and H that begins the precipitation may differ from cloud types as discussed above. In general, the results are used with caution when comparing to other studies in quantifying $S_o$ since the dominating cloud process and the choices applied in how to calculate parameters involved with Eq. (1) can differ widely (e.g., Duong et al., 2011).





## 3 Results

### 3.1 $S_o$ in Sc and Cu

$S_o$ as a function of H is shown in Fig. 4a for Cu. $S_o$ is calculated from Eq. (1) with $N_d$ and R sampled during the cloud-base level-leg flights. Cloud level-leg flights usually last 7-15 minutes on average, with an aircraft speed of 50-60 m s$^{-1}$. In Fig. 4a, $S_o$ during BACEX is about 0 for the clouds shallower than 400 m, then it slightly increases to 0.2-0.3 for 500 < H < 700 m. For H > ~700 m, $S_o$ begins to increase rapidly with increasing H and peaks near H~1400 m with the $S_0$ ~ 1.6. After that, $S_o$ starts to decrease as H increases. The $S_o$ trend during KWACEX follows that from BACEX, especially in the thicker cloud regime where the majority of KWACEX data were sampled. Only four flights were available for KWACEX data analysis and no data were available between 800 m and 1500 m that satisfied the data analysis criteria. Therefore, $S_o$ may peak somewhere H between 600 m and 1700 m by referring to its overall trend.

The qualitative behavior of $S_o$ for Sc is shown in Fig. 4b. $S_o$ during E-PEACE shows H-dependent $S_o$ patterns, which are similar to those from BACEX. In the small H regime (H < 240 m), $S_o$ is almost constant in $S_o$ ~0.2. For H > ~240 m, $S_o$ increases gradually with increasing H and peaks at $S_o$ ~1.0 around H values between 350 m and 400 m. After that, $S_o$ decreases with increasing H. Figure 4b further shows that the overall pattern of $S_o$ is similar regardless of how the cloud bases were determined, although the H at which $S_o$ peaks changes slightly.

During VOCALS, $S_o$ increases with increasing H, from $S_o$ ~0.1 near 170 m to $S_o$ ~0.5 near 300 m. Then, a minimum $S_o$ value is shown near H ~ 640 m. The negative values of $S_o$ in the largest H regime possibly result from either, or both, uncertainties in the $S_o$ estimation in that category or/and from the macrophysical properties that affect the precipitation other than LWP or H in that category. The data used to calculate that point stemmed from one day (1 Nov., Table 1), and further, only one sounding was made during the day. Consequently, it is possible that the negative value of $S_o$ was due to the uncertainty in H, if the H varied substantially during the cloud-base level-leg flight on the day while $S_o$ was calculated with a daily mean H. However, it is also possible that the negative value of $S_o$ in the thicker clouds (i.e., high LWP) was due to the other factors that control the precipitation such as turbulence (Baker, 1993; Ayala et al., 2008), stronger updrafts due to the latent heat release from the precipitation (Rosenfeld, 2008), or increased GCCN in a high $N_d$ environment (e.g., Jung et al., 2015; Terai et al., 2015). The negative value of $S_o$ in the thicker cloud regime is also found in the CAM5 GCM simulation with an excessive accretion rates (e.g., Fig. 7 in Gettelman et al., 2013).

The failure of the cloud radar during VOCALS was responsible for the small resolvable ranges of H that led to only four H groups (Table 1). Additionally, no data was available for H between 350 m and 600 m (Fig. 3), and thus, it is possible that $S_o$ peaks anywhere between H values of 300 m and 600 m. The results of VOCALS clearly show the disadvantage of no cloud radar (i.e., high resolution of LWP or H) for the $S_o$ estimates.



Figure 4 further shows that, in general, $S_o$ tends to be overestimated as the size of data samples (equivalently, temporal and/or spatial resolution of sampling) decreases contrast grey squares to the circles for both Sc and Cu, where $S_o$ of grey squares are analogous (but not equivalent) to larger averaging lengths to circles. The results are similar to Terai et al. (2012), showing high values of $S_o$ when larger averaging length scales are used (see 5 km versus 20 km in their Fig. 7). The

higher $S_o$ values, compared with lower values of $S_o$ that are calculated by using all available data points, probably show the impacts of meteorology on $S_o$ within the fixed H, because the cloud data points close to each other with similar H are more likely to experience the same meteorology.

## 3.2 Autoconversion and Accretion Process in VOCALS-REx and E-PEACE

For cloud droplets to become raindrops (typical diameters of cloud droplets and drizzle drops are about 20 and 200 µm,

respectively (Rogers and Yau, 1989)), they have to increase in size significantly by the collision-coalescence process (autoconversion and accretion). Here autoconversion refers to the precipitation process caused by interactions between cloud droplets. That being said, faster-falling large cloud droplets collect smaller cloud droplets in their paths as they fall through a cloud and grow larger; the accretion process refers to the precipitation process caused by precipitation embryos that collect cloud droplets. In the intermediate LWP regime where $S_o$ increases with LWP or H (ascending branch of $S_o$) the auto-

conversion process dominates. On the other hand, in the high LWP regime where $S_o$ decreases with LWP or H (descending branch of $S_o$) the accretion process dominates (Feingold and Siebert, 2009; Feingold et al., 2013). The transition from the dominance of autoconversion to accretion is reported to occur when $D_e$ exceeds ~ 28 µm, and has been used as a rain initiation threshold in Sc (e.g., Rosenfeld et al., 2012). Jung et al. (2015) also showed that the precipitation embryos appeared (and warm rain initiated) when the mean droplets diameters were slightly less than 30 µm from the salt seeding

experiments during E-PEACE, in the NEP Sc decks (e.g., see Table 3, Fig. 6a, and Fig. 7 in their study). Clouds during VOCALS consisted of numerous small droplets (D < 15 µm in Fig. 5a), which primarily are involved with the autoconversion process except for one flight (D ~37 µm, RF09, Nov. 1). Figure 5a also shows that the size of droplets increases with increasing height, and thus, the overall cloud droplet sizes are slightly larger in the mid-cloud level (red) than those in the cloud-base level (black). Compared with VOCALS Sc decks, Fig. 6 shows that E-PEACE Sc clouds are

composed of larger-sized droplets.

Feingold et al. (2013) showed that time available for collision-coalescence $t_c$ influences the value of $S_o$. That being said, an increase in $t_c$ shifts the balance of rain production from autoconversion to accretion with all else (e.g., LWP and/or H) being equal. Further, they showed that radar reflectivity Z is a good indicator of $t_c$: higher Z coincides with longer $t_c$. To examine how or whether $t_c$ relates to the discrepancy of $S_o$ responses in Sc between previous and current studies, Z with

height is shown in Fig. 5b. Fig. 5c is the same as Fig. 5b, but H is used as y-ordinate. Figure 5c essentially shows that Z increases as cloud deepens, indicating a longer $t_c$ in the thicker clouds. Accordingly, accretion process would dominate (or



play major roles) in the thicker cloud regimes, and in turn, $S_o$ estimated from the thicker clouds will show a descending branch of $S_o$ predominantly, whereas $S_o$ estimated from the thinner clouds will show an ascending branch of $S_o$ mainly, which are consistent with the behavior of $S_o$ in Fig. 4. In Fig. 5b-c, overall, Z calculated from the mid-cloud levels (red) are stronger than Z from cloud-base levels (black) (expect for extremely thick or thin clouds), indicating a longer $t_c$ for the

clouds sampled at mid-cloud level compared with those sampled at cloud-base. The reversed reflectivity pattern shown in the extremely deep or shallow clouds (i.e., Z decreases with height) could be related to a different stage of cloud-life time (e.g., growth or dissipating). But it is not discussed here. Since longer $t_c$ shifts the balance of rain production from autoconversion to accretion, the accretion process would be dominated in the mid-levels, and thus, a descending branch of $S_o$ would be apparent in the clouds sampled at mid-level compared with clouds sampled at cloud-bases. In contrast, the ascending branch

of $S_o$ (equivalently, autoconversion process) would appear predominantly in the clouds sampled close to cloud bases. The effect of the location of in-cloud sampling (mid-cloud versus cloud-base) on the qualitative behavior of $S_o$ will be discussed in the following section by comparing current results with those of Terai et al. (2012).

## 4 Discussion

This study shows the consistent behavior of $S_o$ as a function of a key macrophysical cloud property regardless of cloud type;

i.e., $S_o$ increases with increasing H (ascending branch) and peaks at intermediate H before $S_o$ decreases with H (descending branch) in both Sc and Cu (Fig. 4). The results from marine cumulus clouds (BACEX and KWACEX) are consistent with previous modeling and observational studies of warm cumulus clouds (Sorooshian et al., 2009, 2010; Jiang et al., 2010; Duong et al., 2011; Feingold and Siebert, 2009; Feingold et al., 2013). However, $S_o$ estimated from marine stratocumulus clouds (E-PEACE and VOCALS) are inconsistent with previous in-situ observations of warm stratocumulus clouds (Terai et

al., 2012), but are consistent with previous satellite observations of weakly precipitating Sc (Sorooshian et al., 2010), global climate model simulation (Gettelman et al., 2013), and box and parcel model studies (Feingold et al., 2013) of Sc.

       Possible reasons for why the current results differ from those in previous studies of Sc are discussed here by comparing results from the Terai et al. (2012) study. Although we compare our results with those of Terai et al. (2012), the issues discussed here apply to any results that used the same methods or data analysis as discussed here. The inconsistent

behaviors of $S_o$ between our study and theirs may be due to a number of factors. First, Terai et al. (2012) used R = 0.14 mm day$^{-1}$ as a minimum R threshold to estimate $S_o$ where 0.14 mm day$^{-1}$ corresponds to -15 dBz from the Z-R relationship that they used (R=2.01Z$^{0.77}$ from Comstock et al., 2004). Note that not all of the data shown in Fig. 1 in Terai et al. (2012) are used for the $S_o$ calculation in their study. This R threshold is possibly too high to capture the autoconversion processes that occur in lightly precipitating clouds or clouds that are not precipitating yet but ready to precipitate such as clouds sampled

during VOCALS TO flights. As a result, the high value of minimum R threshold captures the accretion process only (or





predominantly), which may contribute to the descending branch of $S_o$ in their study. As an example, this R threshold rejects all the data in Fig. 2a (VOCALS TO flights) except for one day (RF09, Nov. 1) when the mean effective diameter is about 37 µm and the accretion process dominates for the day. However, it should be also noted that the R threshold had little effect on $S_o$ estimates and its qualitative behavior for E-PEACE (Fig. A2) because the overall $D_e$ during E-PEACE was larger than

that during VOCALS (Fig. 5a versus Fig. 6). In turn, the higher R threshold (which is proportional to $\sim D^4$) did not alter the overall characteristics of the E-PEACE dataset. Nonetheless, it is clear that the choice of minimum R threshold can change the overall character of the dataset used to calculate $S_o$, which is evident in the VOCALS TO flight data. $S_o$ is designed to show the impact of aerosols on precipitation; as aerosol increases, smaller sizes of numerous droplets form, and those droplets suppress the collision-coalescence process, and in turn, precipitation. Therefore, to study the extent that aerosols

suppress precipitation, it would be more reasonable to cover non-precipitating to precipitating clouds (i.e., from lower R to higher R) that include both autoconversion and accretion processes. It is also noted that the framework of precipitation susceptibility is to measure the impact of aerosol perturbations on the precipitation suppression, and thus, the concept of $S_o$ may not adequately apply to the clouds that are already heavily precipitating since the accretion process has little dependence on $N_d$. In addition to decreasing the LWP, the precipitation itself can rainout the aerosols and results in lower $N_d$.

Second, the in-cloud data (R and $N_d$) used in Terai et al. (2012) were mainly obtained from the mid-cloud levels, whereas data used in the current study were obtained from the cloud-base heights. While the VOCALS C-130 flights consist of one in-cloud level-leg (mid-cloud level in most cases), TO flights consist of 2-3 in-cloud level-leg flights that include cloud-base, mid-cloud, and cloud-top level-legs. Cloud data sampled from mid-cloud levels reflect enhanced accretion processes if $D_e$ increases with heights (as discussed in 3.2). In fact, Painemal and Zuidema (2011) showed that $D_e$ increased

with heights in the SEP Sc decks during VOCALS, which is based on C-130 measurements. The increase in $D_e$ with height (e.g., Fig. 5 of Painemal and Zuidema, 2011) is consistent with the $D_e$ distribution shown from the TO flight (Fig. 5a). Therefore, the enhanced (major) accretion process, which appears as a descending branch of $S_o$ predominantly, is expected in the mid-cloud levels compared with that from the cloud-bases (Sect. 3.2). Indeed, Gettelman et al. (2013) showed that the accretion process dominated during VOCALS C-130 flights; the accretion to autoconversion ratio was above 1 for all LWP

ranges during VOCALS observation (e.g., Fig. 5a in their studies).

Third, the overall high values shown in Terai et al. (2012) ($S_o$ begins with around 3 near H~50 m and ends with $S_o$ ~0.8 near H~500 m) may reflect the effects of wet scavenging (Fig. 7a; see also Duong et al., 2011), especially by considering that the drizzle intensity and frequency in SEP Sc decks tended to increase westward from the coast (e.g., Bretherton et al., 2010), and their dataset included several Pockets of Open Cells (POCs) with strong precipitation (personal

communication). We also noted that $S_o$ calculated from the 13 E-PEACE flights was about 0.62. However, $S_o$ calculated from 12 E-PEACE flights that excluded one rainy day was about 0.42, which is consistent with larger $S_o$ in the presence of (heavy) precipitation possibly due to the wet scavenging. Consistently, $S_o$ values calculated from 9 BACEX flights (Cu),





which excluded three heavy precipitation cases, were also shifted to lower values than those estimated from the entire 13 flights (not shown).

Fourth, Terai et al. (2012) used column-maximum Z and then converted the Z to R by using a Z-R relationship when cloud-base was not determined from the lidar due to the attenuation by the heavy precipitation. This procedure can overestimate precipitation for a given $N_d$. If the procedure (i.e., overestimates of R) happens in a low $N_d$ regime (left half of the dotted line in Fig. 7b), the steeper slope (i.e., higher $S_o$) would be obtained (Fig. 7b). If the procedure happens in a high $N_d$ regime, the lower slope would be attained (Fig. 7c). Based on Fig. 1 of their study, the former scenario (Fig. 7b) would occur, resulting in higher $S_o$ than expected.

Fifth, the Z-R relationship that Terai et al. (2012) used (R=2.01Z$^{0.77}$, followed Comstock et al. (2004)'s Z=25R$^{1.3}$) was derived from Sc that was combined near Peru with off the coast of Mexico (ship measurement). The Sc during VOCALS C-130 flights may have a different microphysical process from which the original Z-R relationship was derived. The microphysical processes are responsible for the formation of DSD, and the variability of DSD determines the theoretical limit of precipitation accuracy by radar via Z-R relationship. That being said, changes in DSD imply different Z-R relationships. The DSD variability (e.g., day to day, within a day, between physical processes and within a physical process) causes about 30-50 % of errors in R estimates with a single Z-R relationship (e.g., see Lee and Zawadzki, 2005 and references therein). Besides, the Comstock et al. (2004) Z-R relationship was derived from drop-sizes ranged from 2 µm to 800 µm in diameter (for drops larger than 800 µm, extrapolation was used). The Sc from VOCALS C-130 flights included several POCs, while the clouds that the Z-R relationship was derived were characterized by persistent Sc, sometimes continuous and other times broken with intermittent drizzle throughout. Therefore, using the Z-R relationship of Comstock et al. (2004) may result in some additional uncertainties in R estimates in Terai et al. (2012) as the error of Z-R relationship becomes larger in the bigger drop sizes (Z and R are proportional to $\sim D^6$ and $\sim D^4$, respectively). Further, applying a Z-R relationship to W-band (3 mm) radar returns is not valid if there are any droplets greater than 1 mm since non-Rayleigh scattering (Mie effects) can dominate the radar reflectivity. Note that the Terai et al. (2012) R retrievals were made with a W-band radar. The errors in R estimates with a single Z-R relationship, however, may not critically affect the differences in $S_o$ between studies as $S_o$ metric (Eq. 1) is less sensitive to data uncertainty by using the logarithmic form of the data. Nevertheless, using a Z-R relationship is not an ideal, unless there are no alternatives and/or the microphysical processes are the same in the regions where the relationship is derived and where the relationship is applied.

Sixth, the assumption that Terai et al. (2012) made for the linear relationship between sub-cloud aerosol concentrations and cloud-base $N_d$ may also contribute to the differences. According to Jung (2012 in Fig. 4.5), the linear relationship between sub-cloud aerosols and cloud-base $N_d$ is well established only in the updraft regime, and the linearity gets stronger as a function of updraft velocity, although these results are shown for the marine shallow cumulus clouds. Similarly, using the aerosol proxy from the satellite data for the $S_o$ calculation also needs caution. Jung et al. (2016) showed





that Aerosol optical depth (AOD) is not always a good indicator of the sub-cloud layer aerosols especially when the fine particles from long-distance continental pollution plumes reside above the boundary layer (e.g., Fig 4-5 their study). Mann et al. (2014) also assumed the linearity between cloud-base $N_d$ and 10 m CCN (at 0.55 % super-saturation) for the $S_o$ calculation and showed the same decreasing behavior of $S_o$ with LWP as Terei et al. (2012).

Lastly, to estimate $S_o$ for a given H and LWP interval, Terai et al. (2012) used 30 % of the highest and the lowest $N_d$ and R data on the log ($N_d$) and –log(R) diagram instead of using all available data. This method possibly could affect/change the slope (i.e., $S_o$) more readily (but not necessarily) than the way using all the available data.

**5 Conclusions**

The suppression of precipitation due to the enhanced aerosol concentrations ($N_a$) is a general feature of warm clouds. In this
study we examined precipitation susceptibility $S_o$ in marine low clouds by using in situ data obtained from four field campaigns with similar datasets; two of them focused on marine stratocumulus (Sc), and two targeted shallow cumulus (Cu) clouds. This study shows that the maximum values of $S_o$ are ~1.0 for Sc and ~1.5 for Cu, which are less than the values of $S_o$ of ~2.0 that climate models tend to use for the value of –β in Eq. 2. This study is the first to show with airborne data that for both Sc and Cu, $S_o$ increases with increasing cloud thickness H and peaks at an intermediate H, before decreasing. The
results are consistent with previous studies of warm cumulus clouds, but inconsistent with those of warm marine stratocumulus clouds in-situ observations.

We suggest several possible reasons for why these results differ from those in previous studies of Sc. For example, by comparing with in-situ measurements of Terai et al. (2012). The sources of these uncertainties include the following: (i) high minimum R threshold, (ii) the location of in-cloud sampling (mid-cloud versus cloud-base), (iii) wet scavenging effects
(caused the overall high values of $S_o$), (iv) the use of maximum column Z to convert R under heavy rain conditions, (v) the use of Z-R relationship for the R estimates, (vi) the linearity assumption between sub-cloud aerosols and cloud-base $N_d$, and (vii) the use of partial dataset. Most of these reasons are related with data analysis (except for wet scavenging), but some of them are possibly co-involved with physics (e.g., first and second).

We also note that Z increases with height that is consistent with the H-dependent behavior of $S_o$ that suggests the
predominance of autoconversion process (predominance of ascending branch of $S_o$) in the small H regime, and the dominance of accretion process (predominance of descending branch) in the large H regime. We also note that the details of $N_d$ and R thresholds (Appendix Fig. 1) or how the cloud base is determined have little effect on both $S_o$ values and the qualitative H-dependent behavior (Fig. 4); however, the robust behavior of $S_o$ was because the chosen thresholds did not change the overall character of the dataset. Here we emphasize and caution that the choice of the threshold for the data
analysis because the chosen threshold possibly can alter the character of the dataset that are used to calculate $S_o$ by



subsampling the data. For example, if a high value of the minimum R threshold is chosen in a dataset where the majority of data have low precipitation (e.g., VOCALS TO flights, Fig. 3a) and/or in the bimodal population of precipitation, the threshold would, by chance, eliminate/reduce the autoconversion process associated dataset; and in turn, would show the accretion predominated behavior of $S_o$. The VOCALS C-130 flight datasets are likely dominated by the accretion process

occurring naturally (areas in the POCs), by the adapted flight strategy (mid-level cloud sampling), and by the choice of high R thresholds.

The values of $S_o$ in this study were calculated from in-situ measurements, and thus, no issues associated with the retrieval (e.g., satellite data), empirical relationships (e.g., Z-R relationship), and assumptions (e.g., linearity between sub-cloud aerosols and cloud-base $N_d$) encountered for the calculation of $S_o$. Further, we calculated $S_o$ separately for Cu and Sc to

avoid any possible issues that may arise from mixing different types of clouds (Sect. 2.4). The results, however, should be used with caution when comparing to other studies in quantifying $S_o$ as the dominating cloud process and the choices applied to calculate the parameters in $S_o$ estimates (Eq. 1) can differ widely.

The results of this work motivate further future studies examining the same relationships with a more direct measurement of cloud depth using a cloud radar and/or LWP using a microwave radiometer, in addition to the

instruments/sensors that measure/retrieve R and $N_d$ (Na is also desirable). For the flight strategy, in-cloud level legs at multiple altitudes (cloud-base, mid-cloud and cloud-top) with one sub-cloud level-leg would be ideal to calculate $S_o$ and compare with other studies where $S_o$ is calculated with cloud-base or vertically integrated variables. A level-legs near the ocean surface and sounding(s) to examine the background thermodynamic structures on a given day are also recommended.

The precipitation susceptibility in this study, quantified by the changes in precipitation rate to the changes in cloud

droplet concentrations in the cloud base, showed that R is most susceptible for clouds of medium-deep depth, such as H ~380 m for Sc of which H varies between 100-450 m, and H~1200-1400 m for Cu that H ranges from 200-1600 m. However, R is less susceptible to $N_d$ in both shallow non-precipitating and deep heavily precipitating cloud regimes for both Sc and Cu. The inconsistent behaviors of $S_o$ for the stratocumulus clouds between the current and previous studies are partly attributed to the predominant accretion process in the study area along with some assumptions and thresholds applied to the

data analyses in the previous studies. To capture the characteristic features of the suppression of precipitation rate with aerosol loading, the lower R minimum threshold is desirable to use. Otherwise, the data will be skewed more to conditions where accretion dominates over autoconversion. Further studies on which range of H (or LWP) is most susceptible to precipitation rate would advance our understanding of aerosol impacts on precipitation.





## Appendix A: Sensitivity of R and $N_d$ thresholds to $S_o$ estimates

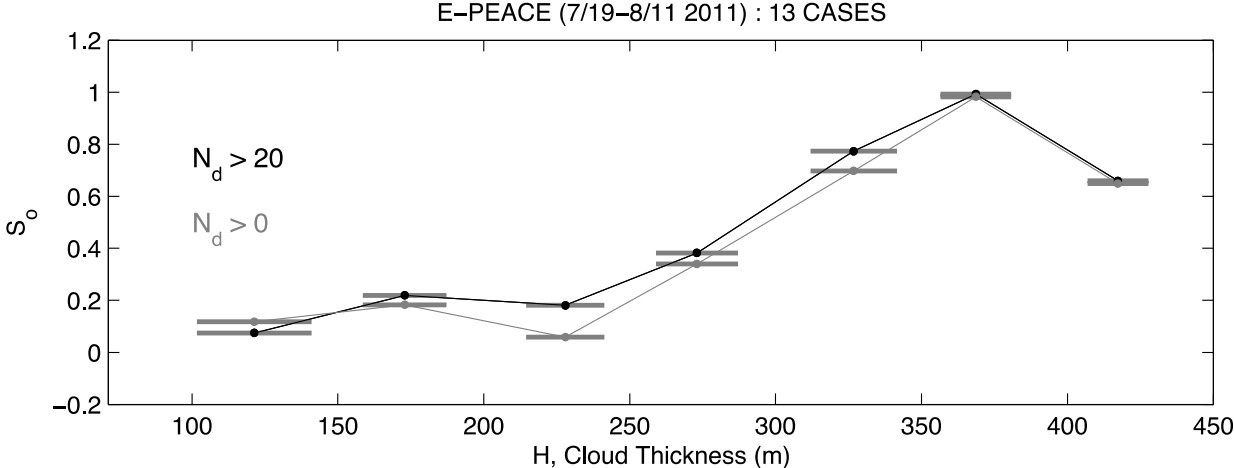

**Figure A1:** The sensitivity of $S_o$ to the $N_d$ threshold value. One standard deviation of mean thickness for a given H interval is shown as horizontal bars.



**Figure A2:** The sensitivity of $S_o$ to the R threshold value. One standard deviation of mean thickness for the given H interval is shown as horizontal bars.



**Table A1**. Table of acronyms and symbols.

| Acronym | Expression |
| --- | --- |
| BACEX | Barbados Aerosol Cloud Experiment |
| CAS | Cloud Aerosol Spectrometer |
| CIP | Cloud Imaging Probe |
| Cu | (Shallow marine) Cumulus (cloud) |
| DSD | Drop Size Distribution |
| E-PEACE | Eastern Pacific Emitted Aerosol Cloud Experiment |
| H | Cloud thickness |
| KWACEX | Key West Aerosol Cloud Experiment |
| LCL | Lifting Condensation Level |
| LWC | Liquid Water Content |
| LWP | Liquid water path |
| $N_d$ | Cloud droplet number concentration |
| PCASP | Passive Cavity Aerosol Spectrometer Probe |
| POCs | Pockets of Open Cells |
| R | Rainfall (Precipitation) Rate |
| Sc | Stratocumulus (clouds) |
| $S_o$ | Precipitation susceptibility |
| TO | Twin Otter |
| VOCALS-REx | VAMOS Ocean-Cloud-Atmosphere-Land Study-Regional Experiment |
| Z | Radar reflectivity |





**Table A2**. H interval and number of data points used in Fig. 4 for each field study.

| E-PEACE, H (m) | H < 130 | 130-160 | 160-190 | 190-220 | 220-250 | 250-280 | 280-310 | 310-340 | 340-370 | 370-400 | 400-430 | H>430 |
|---|---|---|---|---|---|---|---|---|---|---|---|---|
| E-PEACE, # of data points (cb-mean) | 537 | 499 (99%) | 444 (99%) | 508 (99%) | 886 (99%) | 781 (99%) | 606 (99%) | 610 (99%) | 821 (99%) | 373 (99%) | 97 (99%) | 10 |
| E-PEACE (cb-local) | 530 | 482 (99%) | 582 (99%) | 683 (99%) | 726 (99%) | 474 (99%) | 497 (99%) | 525 (99%) | 602 (99%) | 758 (99%) | 236 (99%) | 77 (99%) |
| E-PEACE (cb-lcl) | 514 (< 65%) | 379 (80%) | 341 (90%) | 255 (85%) | 489 (99%) | 823 (99%) | 627 (99%) | 621 (99%) | 670 (99%) | 389 (99%) | 283 (< 65%) | 781 (80%) |
| VOCALS H (m) | Group1 170±27 | Group 2 225±46 | Group 3 307±24 | Group 4 641±201 | - | - | - | - | - | - | - | - |
| VOCALS # of data points | 1113 (80%) | 1280 (99%) | 833 (99%) | 224 (95%) | - | - | - | - | - | - | - | - |
| BACEX H (m) | 0-250 | 250-500 | 500-600 | 600-800 | 800-1000 | 1000-1250 | 1250-1500 | H>1500 | - | - | - | - |
| BACEX # of data points | 23 | 86 (90%) | 40 (90%) | 87 (70%) | 52 (99%) | 38 (99%) | 30 (99%) | 27 (99%) | - | - | - | - |
| KWACEX H (m) | H<1500 m | 1500-1800 | H > 1800 m | - | - | - | - | - | - | - | - | - |
| KWACEX # of data points | 56 (90%) | 32 (99%) | 42 (65%) | - | - | - | - | - | - | - | - | - |

Numbers in parenthesis indicate statistical significance of the linear fits (two-tailed t-test).





**Acknowledgements**

The authors gratefully acknowledge the crews of the CIRPAS Twin Otter for their assistance during these field campaigns. EJ acknowledges Chris Terai for his helpful discussion of the estimate of precipitation susceptibility. This study was funded by ONR Grants N000140810465 and N00014-10-1-0811, and NSF Grant AGS-1008848.

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



**Table 1**. Dates used for this analysis during each experiment.

| No. | VOCALS (Sc) | E-PEACE (Sc) | BACEX (Cu) | KWACEX (Cu) |
|---|---|---|---|---|
| Period | Oct.-Nov., 2008 | July-Aug., 2011 | Mar.-Apr., 2010 | May, 2012 |
| Location | Southeast Pacific Sc decks | Northeast Pacific Sc decks (California coast) | Barbados (Caribbean Sea and North Atlantic) | Key West (Caribbean Sea) |
| RF1 | 10/16 (2) | 7/19 | 3/22 | 5/22 (1st flight) |
| RF2 | 10/18 (3) | 7/21 | 3/23 | 5/22 (2nd flight) |
| RF3 | 10/19 (3) | 7/22 | 3/24 | 5/23 |
| RF4 | 10/21 (1) | 7/23 | 3/25 | 5/24 |
| RF5 | 10/22 (2) | 7/26 | 3/26 | - |
| RF6 | 10/26 (2) | 7/27 | 3/29 | - |
| RF7 | 10/27 (1) | 7/29 | 3/30 | - |
| RF8 | 10/30 (2) | 8/2 | 3/31 | - |
| RF9 | 11/1 (4) | 8/3 | 4/5 | - |
| RF10 | 11/9 (1) | 8/4 | 4/7 | - |
| RF11 | 11/10 (1) | 8/5 | 4/10 | - |
| RF12 | 11/12 (2) | 8/10 | 4/11 | - |
| RF13 | 11/13 (1) | 8/11 | - | - |

*The group number is shown for VOCALS in the parenthesis.

*RF indicates the Research Flight. However, note that RFs from E-PEACE and VOCALS are not the same as RF from Russell et al. (2013) and Zheng et al. (2011), respectively.





**Figure 1:** The geographical location of each field campaign (blue solid). E indicates E-PEACE, K indicates KWACEX, and B shows BACEX. The entire domain of VOCALS-REx is displayed as a solid grey box with domains of C-130 (dashed grey) and TO (solid blue) flights.







**Figure 2:** Scatter diagrams of cloud droplet number concentrations $N_d$ and precipitation R for four field campaigns. Colors indicate cloud thickness H. R increases upward in y ordinate and $N_d$ increase toward the right direction in x abscissa.



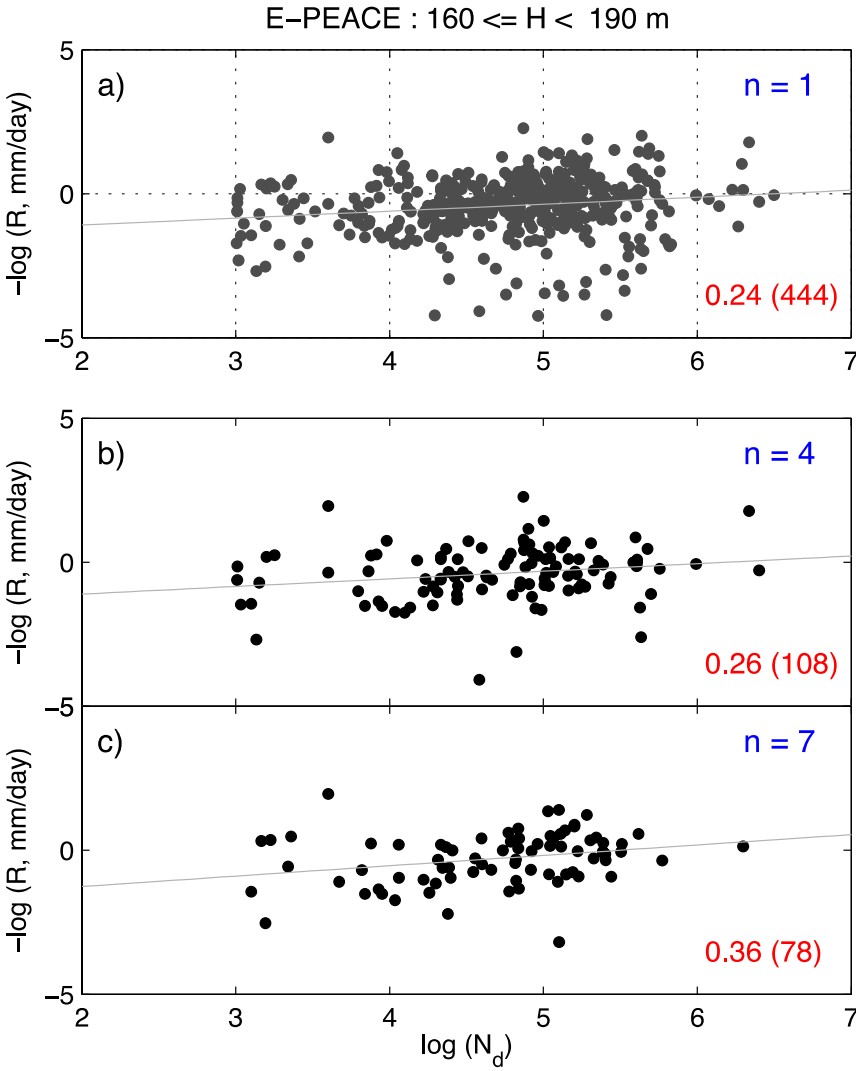

**Figure 3:** Examples of scatterplots used to calculate precipitation susceptibility $S_o$ (i.e., the slope) for E-PEACE. Black dots in each box indicate data points for an H interval between 160 m and 190 m. Numbers on the bottom right (red) indicate the $S_o$ with a total number of data used to calculate $S_o$ inside the parentheses. Numbers on the upper right corner (blue) indicate the interval of sampling size. For example, N=7 in Fig. 3(c) shows data at every 7 seconds (in sequence) were used. R increases downward in y ordinate, and $N_d$ increases toward the right direction in x abscissa.



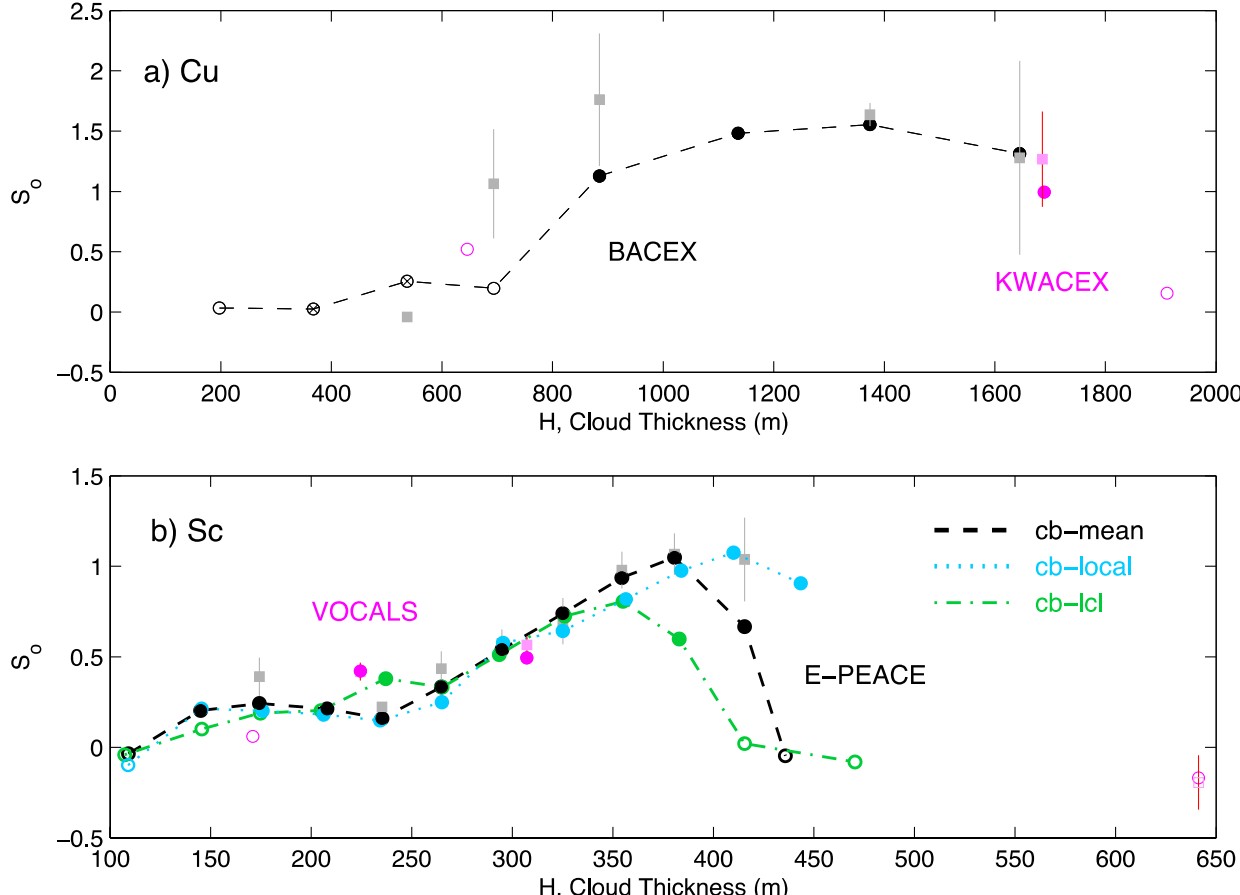

**Figure 4:** Precipitation susceptibility, $S_o$, estimated with aircraft measurements for (a) Cu (12 flights of BACEX and four flights of KWACEX) and (b) Sc (13 flights of E-PEACE and VOCALS-REx). $S_o$ of BACEX and E-PEACE are shown as connected lines since the data covers cloud thickness without gaps. E-PEACE $S_o$ is estimated from (i) the cloud base height, which is identified using LCL (cb-lcl) and from the vertical structures of LWCs (lowest height that the vertical gradient of LWC is the greatest) that ii) the aircraft enters to the cloud deck to conduct the cloud-base level leg flight (cb-local), and (iii) from the averaged cloud-base heights from the nearby soundings and cb-local (cb-mean). $S_o$ that is calculated with the subsets of data points (n=2 to 10 seconds in sequence) is shown as grey (mean) with vertical bars (±1σ). Filled circles and squares are statistically significant at 99 % confidence level. The number of data points used for $S_o$ estimates and its statistical significance is shown in Table A2.



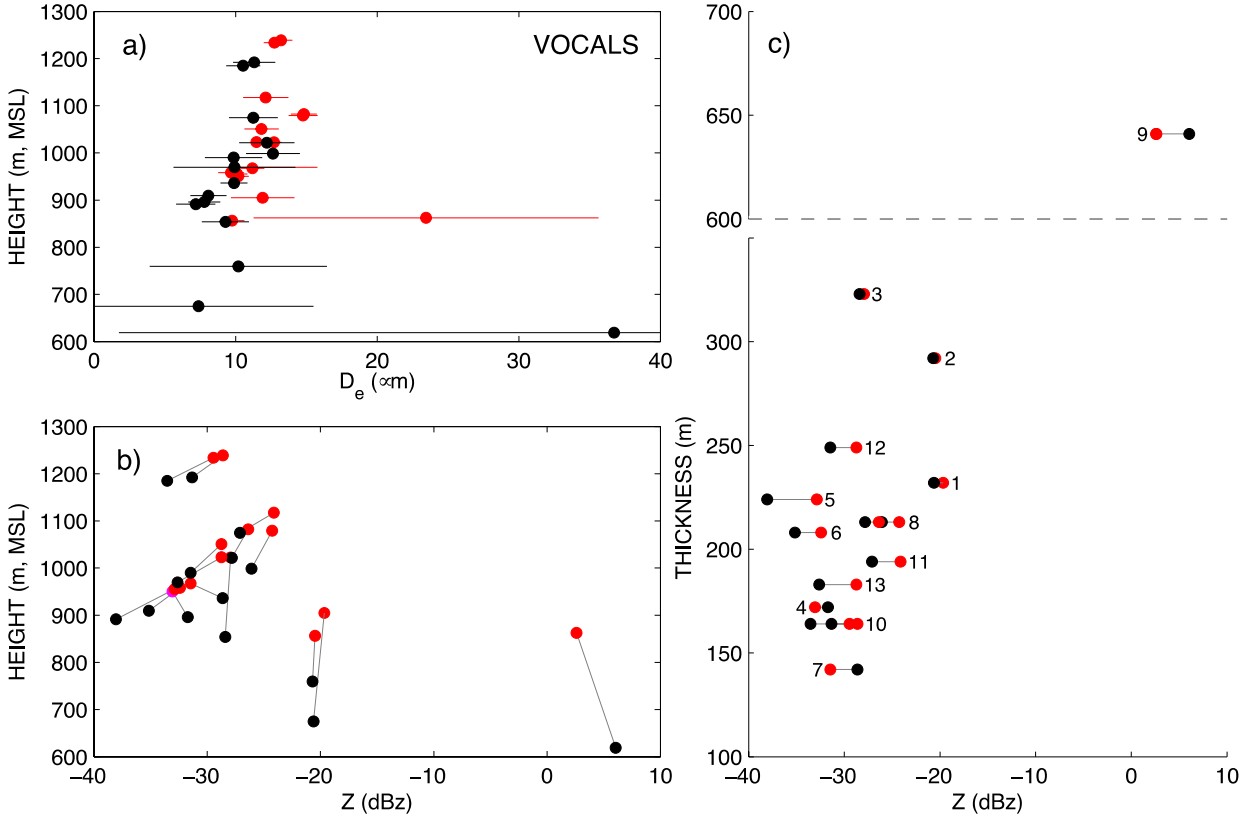

**Figure 5:** Distribution of (a) effective diameters and (b) radar reflectivity of the Southeast Pacific Sc decks (VOCALS-Rex), sampled from mid-cloud (red) and cloud-base (black) levels. Radar reflectivity with cloud depth is shown in Fig. 5c. The pairs of clouds are connected with lines in Fig. 5b-c, and numbers in Fig. 5c indicate research flight (RF) numbers (see Table 1).




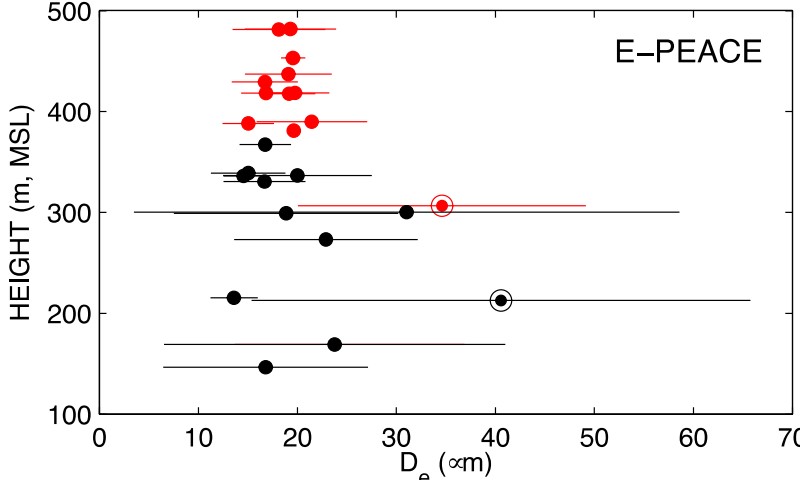

**Figure 6:** Effective diameters of Northeast Pacific Sc decks (E-PEACE) on the mid-cloud (red) and cloud-base (black) levels. Cloud droplets on 11 August are shown as double circles.



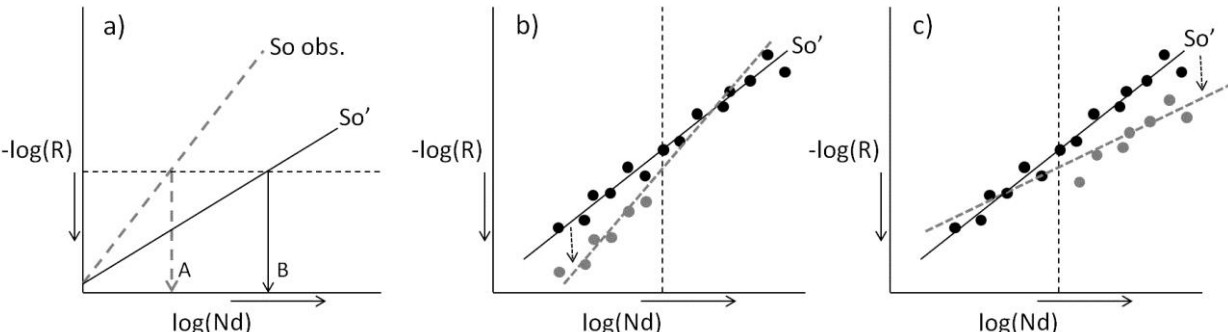

**Figure 7:** A visual description of (a) the effect of wet scavenging and (b-c) the impact of an increase in rainfall rate for a given range of $N_d$ on the estimate of $S_o$. The solid line represents true (expected) $S_o$, whereas the dashed line indicates observed (or responded) $S_o$. The black filled circles in (b-c) indicate the initial (or actual) data and the grey filled circles indicate newly adjusted (responded) data accordingly to the scenario. R increases downward in y ordinate and $N_d$ increase toward the right direction in x abscissa.