# Peer review of "Precipitation Susceptibility in Marine Stratocumulus and Shallow Cumulus from Airborne Measurements"

_Atmospheric Chemistry and Physics, 2016_

## Referee Comment (RC1) · Anonymous Referee #1 · 18 Apr 2016

Review of Jung et al. "Precipitation Susceptibility in Marine Stratocumulus and Shallow Cumulus from Airborne Measurements"

General comments

This study examines the precipitation susceptibility metric using consistent measurements from a variety of field campaigns to ask whether the qualitative behavior of the susceptibility varies with cloud type. Whereas previous studies appear to disagree on whether susceptibility should increase and then decrease with cloud thickness or whether they should decrease monotonically, different retrieval methods were used in these previous studies, so it has not been clear whether the differences are due to cloud types, retrieval methods, or analysis methods. This study does not suffer from

many of the same issues, because the measurements are made from the same aircraft, using the same instruments and sampling strategy.

The authors show that the precipitation susceptibility increases, and then decreases, regardless of whether cumulus or stratocumulus clouds are examined. After presenting their susceptibility estimates, they provide possible explanations for the why the results of Terai et al. (2012) do not capture the increase in susceptibility at lower cloud thicknesses.

The study addresses an existing disagreement in the qualitative behavior amongst precipitation susceptibility estimates and provides valuable observations to add to the existing observed estimates and to try to reconcile the disagreements. However, there some issues that need to be addressed before I recommend publication. In particular, issues of Nd and H covariability and the statistical independence of the 1-second data should be addressed.

Major comments 1) In the study, it appears that a good amount of consecutive data are included in the log(Nd) vs. log(R) regression slopes. Given that N and R estimates from every second are used, there is the possibility that covariability between N and H at those smaller spatial scales might affect S0 estimates. For example, even within the same cloud thickness bin, the N and H can covary in a flight leg due to updraft/downdraft organization. In other words, where there are stronger updrafts, Nd will likely be higher, as well as H. This can impact S0, because H also controls R. Therefore, I would like the authors to examine the extent to which the covariability between N and H exist and how they might affect S0. Do data need to be averaged over longer timesteps to reduce the covariability?

2) Similarly, I would like to see the authors demonstrate whether 1-second of data (N, R) is statistically independent from one another. For example, Leith et al. (1973) provide a method to determine the e-folding time scale, which will help determine whether using the 1-second data is indeed appropriate.
Leith, C. E. (1973), The Standard Error of Time-Average Estimates of Climatic Means, J. Appl. Meteorl., 12, 1066–1069.

3) Whenever a slope is calculated, the statistical uncertainties should also be reported, since the relationship does not appear to be linear in many of the cases (Fig. 2).

4) Possible explanations are presented as to why the results in this study disagree from what is presented in Terai et al. (2012) but are not tested. I believe some of the issues can be tested using the data analyzed this study. For example, the authors can test whether the method used in Terai et al. (2012) gives a different behavior than when a linear regression is used.

5) Many times, in comparing with the results of Terai et al. (2012), their SR is compared with the S0 in this study. Is this the right comparison to make? Or should SI be compared with S0 in this study, since SI captures the effect of aerosols on measureable precipitation rates.

Minor and specific comments

P1 L23: "R" and other variables (e.g. Nd) should be italicized throughout the manuscript

P2 L26-27: "S0 is insensitive to aerosol perturbations where clouds do not precipitate": S0 should be undefined where clouds do not precipitate, not zero.

P2 L30: Please write out what VOCALS stands for.

P3 L6: (and subsequent uses) Replace "GCCM" with "GCM"? If it is supposed to be GCCM, please state what it is an acronym for.

P3 L12: Please define the acronym ARM.

P3 L16: For completeness, at some point in the paragraph the study of Hill et al. 2015 should be mentioned. There are a number of other instances throughout the study where comparison with results of Hill et al. (2015) would also be good to make.

[Figure]

Hill, A. A., B. J. Shipway, and I. A. Boutle (2015), How sensitive are aerosol-precipitation interactions to the warm rain representation?, J. Adv. Model. Earth Syst., 7, 987–1004, doi:10.1002/2014MS000422.

P4 L12: "LCL varied little for Cu" Can the authors attach some numbers to this statement?

P4 L18-26: How long were these cloud base level flights? In other words, over what length scales are cloud thicknesses assumed to be constant, and is this a good approximation?

P4 L28-29: Is using 1 second data appropriate? Given the sampling volume rate and scarcity of drizzle drops I wonder how statistically robust the R retrievals are. Just based on counting statistics, what are the measurement uncertainties in R? What is the theoretical minimum threshold on R given the sampling rate of 1 second?

P5 L3-6: Same question can be applied to z.

P6 L6: What are the 95 or 99% confidence intervals on this estimate? The scatter in Fig. 2 appears rather large.

P7 L2: A measure of the uncertainty will be helpful here as well.

P7 L8: How were the H intervals chosen? Do the results vary with larger or smaller H bins?

P7 L14-15: "We noted that S0 tended …. (not shown)." Why do the authors think that when the Nd variations are small, S0 is high? Are N and h covariations leading to unrealistically high S0? Do uncertainties in the slopes increase in this case?

P7 L16: "if dlog(Nd) spans at least 2.2" Is the natural log used here? If so, what is the threshold of max(Nd)/min(Nd) used here? Perhaps around 10?

P7 L16-17: "exceeds six for a given H" – this seems like a very small sample size for calculating slopes. Uncertainties in the slopes should be shown.

P7 L20: "statistically significant at the 99% confidence level" - I suspect this means statistically significant with a comparison with a slope of 0. This is only the case if the each 1 second of data is independent of another. The authors need to demonstrate that this is the case, perhaps using the method of Leith (1973) or Bretherton (1999).

Leith, C. E. (1973), The Standard Error of Time-Average Estimates of Climatic Means, J. Appl. Meteorl., 12, 1066–1069.

Bretherton, C. S. et al. (1999), The Effective Number of Spatial Degrees of Freedom of a Time-Varying Field, J. Clim., 12,7, 1990-2009.

P 7 L29-30: "S0 tends to be overestimated..." Based on what has been shown so far, it doesn't appear that S0 in necessarily overestimated when a larger 'averaging lengthscale' is used. It can be that S0 is underestimated when every second of data are included.

P7 L32: "H is estimated from the vertical structure of LWC for each day" - If daily mean H is used, then the sub-scale covariance between N and H should be examined, based on the other measurements. To what extent are H and N covarying and how can that potentially affect susceptibility estimates?

P8 L1: "H of 9 Nov. (164+/- 18m)" To what extent is using daily mean H to group data appropriate? What is the true range of H from each day of flight? Are there cases where data from one day could potentially lie in a different bin?

P8 L3: "as the each" – remove "the"

P8 L9: "no data were available between 800 m and 1500 m that satisfied the data analysis" – were there not enough data points that existed in this range to calculate the regressions or was the range of N too small? Could bins have been combined to get an estimate? In Fig. 5, it is difficult to make out much of a trend based on three susceptibility estimates.

P9 L17-19: "The negative values of S0 in the largest... in that category" – based on

the open circle designation (Fig. 4b), it appears that the susceptibility is statistically indistinguishable from zero, so there is no need to explain why it has a negative value.

P9 L20-22: "... if the H varied substantially during the cloud-base level-leg flight on the day which S0 was calculated with a daily mean H." I wonder how the susceptibility estimates from the thinner clouds are similarly affected from the VOCALS flights. Can the authors point to any other data or references, which show that the cloud thickness variability in a flight day are smaller than the variability from flight to flight?

P10 L5-7: "probably show the impacts of meteorology on S0 within the fixed H, because the cloud data points close to each other with similar H are more likely to experience the same meteorology": Although the authors appear to argue that using larger averaging lengths lead to an overestimation of S0, can you not argue that S0 can be underestimated with a shorter averaging length due to covariance between N and H and smaller spatial scales?

P10 L7: The authors have addressed the similarity in the qualitative behavior of susceptibility between Cu and Sc clouds. Can they comment on how they agree in terms of absolute values? And at which thicknesses the peaks occur? Based on previous studies, is there a prior expectation of whether the peak should occur at the same location (H-value)?

P11 L3-5: "... indicating a longer tc for the clouds sampled at mid-cloud level compared with those sampled at cloud-base." - In the developing stages of precipitation, this may be true, but if the drops start to fall out, they will eventually fall through bottom of the cloud, which means they will have a longer tc at the bottom of the cloud. One would expect the cloud base measurements to be a combination of parcels with short tc and with long tc.

P11 L19-20: Include comparison and references to Mann et al., 2014 and Hill et al., 2015.

[Figure]

P11 L27: "Note that not all of the data shown in Fig. 1 in Terai et al. (2012) are used for the S0 calculations in their study." Because their SR included a component coming from Sf, which took into account non-precipitating clouds, all of the data was used to calculate SR. Not all of the data was used for SI.

P12 L1-3: "As an example, this R threshold rejects all the data in Fig. 2a.." How about the H in the two datasets? Are there overlaps? Do the H and R values agree between the two studies?

P12 L15-16: This is not the case. Their reflectivities were mainly taken from cloud-base retrievals and their N was the accumulation aerosol concentration, not the cloud droplet number.

P12 L25: Would the authors summarize the main conclusion of the paragraph? Is it that Terai et al. (2012) examined relationships in mid-cloud level, where accretion rates are high, and therefore, examine only the downward tail of susceptibility?

P13 L4-5: "This procedure can overestimate precipitation for a given Nd" – please elaborate on why this is the case. In many heavily precipitating clouds, the reflectivity is highest at cloud base.

P13 L28-29: Although the use of sub-cloud aerosol concentration to calculate the susceptibility in Terai et al. (2012) might explain some of the differences between the susceptibility in this study and the susceptibility in their study, they did not make the assumption that the sub-cloud aerosol concentrations and cloud-base Nd were linearly related (see their Fig. 2 and the corresponding discussion).

P14 L4: replace "Terei" with "Terai"

P14 L6-7: "This method possibly could affect/change the slope. . ." One way to test this is to apply their method to the data in this study to determine whether susceptibility values using the data in this study are indeed overestimated when using their method.

P14 L24-25: "We also note that Z increases with height that is consistent with the H-

dependent. . ." The Z will also increase with height just from an increase in condensed water that you would get from an adiabatic increase with height.

P15 L26: "the lower R minimum threshold is desirable to use" – what threshold should the minimum be? Should sedimentation of cloud droplets be included? The appendix in Hill et al. 2015 suggests that the asymptotic limit of S0 for small LWP, where 'precipitation' is dominated by cloud sedimentation, is 2/3.

Table 1: In the captions please specify what the numbers in parentheses mean.

Figure 2: What does the dashed line indicate?

Figure 3: Please provide the uncertainties in the slopes.

Figure 4: What do the lighter pink colors indicate? As in my previous comment, the uncertainties in the slopes should be shown.

Figure 5a: What do the horizontal bars indicate?

Figure 6: What do the horizontal bars indicate?

Figure 7a: What do A and B indicate?

---

## Referee Comment (RC2) · Anonymous Referee #2 · 10 May 2016

The authors prepare an analysis of precipitation susceptibility based on in situ measurements from four field campaigns with nearly identical aircraft payloads, two that sampled cumulus clouds and two that sampled stratocumulus. The authors report robust patterns, and advance hypotheses for the trends that they find, as well as possible explanations for differences between their findings and the results of past studies. The methodology for how to best make such calculations is not clearly settled upon, as evidenced by some of the sensitivity tests presented, but details are sufficient for this work to be reproduced. I rate that revisions to address the following comments can make this into a methodologically sound contribution that is suitable for publication. In particular, I think it needs to be taken into account whether 1-s samples are statistically

independent when evaluating sample size. In addition, the authors seem to conflate sample size and horizontal scale dependence, which appears illogical. I also could not follow the arguments about autoconversion versus accretion using this data set, leaving final statements seeming unsupported by the evidence provided. Comments are enumerated below.

1. There should be some guidance on the spatial scale over which equation 1 applies. A reader must assume implicitly based on this work that it is intended to apply to one second of flight time (100 m in horizontal and full vertical column)? Also a GCM grid cell (100 km in horizontal and full vertical column)? Really both identically? Please offer at least brief guidance for the reader in the introduction.

2. Can you pls comment on whether treating $N_d$ as a proxy for $N_a$ has any relevant consequences? For instance, does that proxy give stronger $S_o$ than using $N_a$ owing to a decreasing fraction of aerosol activated with $N_a$ increasing, all else being equal?

3. I recommend revising the text to reflect the fact that not all current climate models use equation 2, such as those with prognostic precipitation species.

4. Should GCCM be GCM throughout?

5. Using 1-second data, there is a big enough sample volume to accurately calculate Z from $dZ/dD$? For instance, can you show evidence that your 1-s sample volume is large enough to produce a smoothly continuous DSD? If everyone except me knows that this is possible, perhaps you can just point to a reference or provide a figure outside of the manuscript.

6. Page 6, line 17: It is stated that figure 2 "essentially shows that as $N_d$ increases, R decreases." I would not jump to that conclusion from that figure. The amount of scatter around the trend in figure 3 demonstrates why. I would recommend leaving this statement out of the introduction to figure 2 and focusing instead on the fact that it shows well the range of R and $N_d$ sampled during each experiment.

7. Page 7, paragraph containing line 25: Can the authors present evidence to indicate that sequential 1-s samples are statistically independent? It seems to me that a methodologically appropriate test of sample size for this study would be to randomly resample the data (if consecutive 1-s points are statistically independent) or else randomly resample the flights used (if they are not).

8. Page 8, lines 9-11: Are the authors suggesting to use one LWP profile for each date for Sc and Cu? Unclear if this statement is limited to Sc.

9. Page 10, first paragraph: This logic is not sound as currently written. First the authors state that So decreases with increasing sample size. There must be a limit to that if the system is well-defined and the significance of the results robustly evaluated, right? Then the authors compare such behavior to that found by others when decreasing the averaging length scale, which is a different issue entirely (see comment 1).

10. 2nd paragraph of section 3.2: I really couldn't follow this paragraph. I would remove section 3.2 and figures 5 and 6 if the point of this paragraph can't be significantly clarified. Stating "But it is not discussed here." furthered this reader's impression that the Z analysis did not really add anything to this study.

11. Page 11, line 23: There is no reason to show a figure such as A2. Simply state that results are insensitive. I would be much more interested to see a clear demonstration of a case where the R threshold is very important. It seems clear to the authors, but is not so clear to me how figure 4 would be affected, for instance.

12. Page 14, line 24: I really did not take away the autoconversion versus accretion behavior. There seemed to be a lot of handwaving in section 3.2. I basically feel that this statement is just not supported by the material shown. I think this needs to be much clarified or else removed.

13. I don't understand the last sentence of the paper. The authors call for more studies

on which range of H is most susceptible to preciptiation rate? This is a study on susceptibility of precipitation rate to Nd. Are the authors suggesting another thing? If the authors meant to further study So as defined here, why are further studies needed? All previous sentences in the last paragraph would indicate that the authors have already shown within which range of H Sc and Cu are most susceptible. Are these results somehow uncertain or incomplete? If that could be clarified and its relevance to the conclusions made here (regarding the general behavior of So in Sc and Cu found; is that uncertain?), I think that would better support this closing argument, if I understand it correctly.

14. So many grammatical errors appear here in a paper with so many capable co-authors that I will not take my time to enumerate them, but merely note that this sometimes impacted my ability to evaluate the work (as in comment 13 above).

15. Please label Nd axis units on figures 2 and 3.

---

## Author Comment (AC1) · 15 Jul 2016

We thank both reviewers for thoughtful suggestions and constructive criticism that have helped us improve our manuscript. Below we provide responses to each reviewer's concerns and suggestions in blue font. Please refer to the attached file (replies to each comment and a revised manuscript with track marked).

Please also note the supplement to this comment:
http://www.atmos-chem-phys-discuss.net/acp-2016-161/acp-2016-161-AC1-supplement.zip

---

## Author Response (AR1)

Review of Jung et al. "Precipitation Susceptibility in Marine Stratocumulus and Shallow Cumulus from Airborne Measurements"

General comments
This study examines the precipitation susceptibility metric using consistent measurements from a variety of field campaigns to ask whether the qualitative behavior of the susceptibility varies with cloud type. Whereas previous studies appear to disagree on whether susceptibility should increase and then decrease with cloud thickness or whether they should decrease monotonically, different retrieval methods were used in these previous studies, so it has not been clear whether the differences are due to cloud types, retrieval methods, or analysis methods.
This study does not suffer from many of the same issues, because the measurements are made from the same aircraft, using the same instruments and sampling strategy.

The authors show that the precipitation susceptibility increases, and then decreases, regardless of whether cumulus or stratocumulus clouds are examined. After presenting their susceptibility estimates, they provide possible explanations for the why the results of Terai et al. (2012) do not capture the increase in susceptibility at lower cloud thicknesses.

The study addresses an existing disagreement in the qualitative behavior amongst precipitation susceptibility estimates and provides valuable observations to add to the existing observed estimates and to try to reconcile the disagreements. However, there some issues that need to be addressed before I recommend publication. In particular, issues of $N_d$ and H covariability and the statistical independence of the 1-second data should be addressed.

Response: As we will elaborate upon below, we have addressed both of these issues. We show that in our datasets, $N_d$ and H do not co-vary and have showed that our results and conclusions are robust using 1-second data in favor of other methods proposed by the reviewers.

Major comments 1) In the study, it appears that a good amount of consecutive data are included in the log(Nd) vs. log(R) regression slopes. Given that N and R estimates from every second are used, there is the possibility that covariability between N and H at those smaller spatial scales might affect $S_0$ estimates. For example, even within the same cloud thickness bin, the N and H can covary in a flight leg due to updraft/downdraft organization. In other words, where there are stronger updrafts, Nd will likely be higher, as well as H. This can impact S0, because H also controls R. Therefore, I would like the authors to examine the extent to which the covariability between N and H exist and how they might affect S0. Do data need to be averaged over longer timesteps to reduce the covariability?

Response: The scatter diagram of $N_d$ and $H$ is shown in Fig. 1 for E-PEACE as an example. There is a weak correlation (or co-variability) between $N_d$ and $H$. The correlation ($r$) between two is about ~0.03, and the covariance is about 0.18. We also show results below for a couple representative individual flights and show that $N_d$ and $H$ do not co-vary all the time.

[Figure]

**Figure 1A**. Scatter diagrams of $N_d$ and $H$ for the ten flights of E-PEACE.

[Figure]

**Figure 1B**. Scatter diagrams of $N_d$ and $H$ for the random individual flights during E-PEACE. Numbers on the upper right corner indicates a covariability between $N_d$ and $H$ for a given flight.

As a result, the data do not need to be averaged over longer time-steps to reduce the co-variability as the co-variability is small. However, we also calculated $S_o$ with data that are averaged over longer time-steps by considering the independence of the 1-second data (shown later).

2) Similarly, I would like to see the authors demonstrate whether 1-second of data (N, R) is statistically independent from one another. For example, Leith et al. (1973) provide a method to determine the e-folding time scale, which will help determine whether using the 1-second data is indeed appropriate.
Leith, C. E. (1973), The Standard Error of Time-Average Estimates of Climatic Means, J. Appl. Meteorl., 12, 1066–1069.

Response: We find that the e-folding time of $N_d$ during E-PEACE varies from about 4-6 minutes to 10 minutes, and an e-folding time of $R$ varies from a few seconds to 1-2 minutes. The e-folding time of $N_d$ within the VOCALS-TO flights varied between 2-6 minutes, and for the cloud-base precipitation was less than (or approximately)

1 minute (less than 3 km, consistent with Terai et al., 2012). In the case of BACEX (Cu), the overall e-folding times were much shorter, varying from 1-2 minutes for $N_d$ and less than 1 minute for $R$. The e-folding times of $N_d$ and $R$ are summarized in Table 1 for VOCALS, E-PEACE and BACEX. KWACEX was not included since there were only four flights.

We calculated $S_o$ with data averaged over the upper bounds of the e-folding time (i.e., e-folding time of $N_d$) for E-PEACE, BACEX and VOCALS flights, and the qualitative behavior of $S_o$ reported with 1-second data is unchanged: $S_o$ increases with $H$ then peaks before it decreases again (Fig. 5 for BACEX and E-PEACE and Fig. 6 for VOCALS. Fig. 6 is shown later). However, it should be noted that the $H$ that $S_o$ peak is shifted toward the lower $H$, which is consistent with the results of Duong et al. (2011). The shift of $H$ to the lower $H$ is substantial in Sc where the overall $H$ is smaller than H of Cu.

[Figure]

**Figure 5**. $S_o$ estimated with aircraft measurements for (a) BACEX (Cu) and (b) E-PEACE (Sc). The 1-second data of individual flights are reduced by averaging over the e-folding time of $N_d$ for each day prior to the calculation.

3) Whenever a slope is calculated, the statistical uncertainties should also be reported, since the relationship does not appear to be linear in many of the cases (Fig. 2).

Response: The slope is calculated from $\ln(N_d)$ and $-\ln(R)$ that are shown in Fig. 3, whereas Fig. 2 shows $N_d$ and $\log(R)$. The linear correlations ($r$) is added in the $\ln(N_d)$ and $-\ln(R)$ diagrams and $r$ and P-values indicating the statistically significant level of confidence for the fitted lines are summarized in Table A2 for given $H$ intervals.

4) Possible explanations are presented as to why the results in this study disagree from what is presented in Terai et al. (2012) but are not tested. I believe some of the issues can be tested using the data analyzed this study. For example, the authors can test whether the method used in Terai et al. (2012) gives a different behavior than when a linear regression is used.

Response: Please refer to the replies to "P14 L6-7" later (tercile log-differencing versus linear regression). We also believe that one of the most fundamental reasons causing the difference comes from the differences in cloud fields between that were sampled during the flights. For example, VOCALS TO sampled cloud fields close to the continent that had high aerosol concentration with weak precipitation. In contrast, VOCAL C-130 flight sampled the cloud fields in the open ocean where the cloud fields consisted of Pockets of Open Cells (in many cases), and more intense and frequent precipitation was observed.  The effect of precipitation on the $S_o$ estimates (and the effect of high R threshold on the $S_o$ estimates) are shown later (Please refer to the replies to #7 for the second reviewer or Figure B2 and section 3.2).

5) Many times, in comparing with the results of Terai et al. (2012), their SR is compared with the $S_0$ in this study. Is this the right comparison to make? Or should SI be compared with $S_0$ in this study, since $S_I$ captures the effect of aerosols on measureable precipitation rates.

Response:
Terai et al. define the $S_o = S_I + S_f$, which corresponds to $S_o$ in current study.

Terai et al. used 10-km segment-averaged cloud data and determined the fraction and intensity of the drizzle in each segment. The segment-mean precipitation rates R is partitioned into the fraction of the cloud columns that are drizzling f, and the mean drizzle rate in that column (drizzle intensity I). Their SI is calculated exclusively for the clouds that produce measurable precipitation, which is set by the R threshold, and their $S_f$ is considered for all clouds.

On the other hand, in the current study, cloud data are included in the analysis if the given precipitation rate is greater than a threshold of 0.001 mm day$^{-1.}$ The low $R$ threshold is intended to include both non-precipitating and precipitating clouds.

Minor and specific comments
P1 L23: "R" and other variables (e.g. Nd) should be italicized throughout the manuscript

Response: Revised as the reviewer suggested.

P2 L26-27: "S0 is insensitive to aerosol perturbations where clouds do not precipitate": S0 should be undefined where clouds do not precipitate, not zero.

Response: The manuscript is revised to clarify the point as follows: At low LWP, not enough water is available with which to initiate rain, and $S_o$ is insensitive to aerosol perturbations

P2 L30: Please write out what VOCALS stands for.

Response: Revised as VAMOS Ocean-Cloud-Atmosphere-Land Study-Regional Experiment (VOCALS-REx).

P3 L6: (and subsequent uses) Replace "GCCM" with "GCM"? If it is supposed to be GCCM, please state what it is an acronym for.

Response: Corrected.

P3 L12: Please define the acronym ARM.

Response: Revised as Atmospheric Radiation Measurement (ARM)

P3 L16: For completeness, at some point in the paragraph the study of Hill et al. 2015 should be mentioned. There are a number of other instances throughout the study where comparison with results of Hill et al. (2015) would also be good to make.

Hill, A. A., B. J. Shipway, and I. A. Boutle (2015), How sensitive are aerosol-precipitation interactions to the warm rain representation?, J. Adv. Model. Earth Syst., 7, 987–1004, doi:10.1002/2014MS000422.

Response: The reference is added in the revised manuscript.

P4 L12: "LCL varied little for Cu" Can the authors attach some numbers to this statement?

Response: Revised as follows: "The LCL varied little for Cu, for example, during the Barbados Aerosol Cloud Experiment (Sect. 2.3), the LCLs were 653.9±146 m on average from the aircraft measurements, which agreed with the two-year LCL climatology in this region (700±150 m) documented in Nuijens et al. (2014)"

P4 L18-26: How long were these cloud base level flights? In other words, over what length scales are cloud thicknesses assumed to be constant, and is this a good approximation?

Response: Cloud-base level flights last about 10-20 minutes. H=CT-CB where H indicates cloud thickness, CT and CB indicate cloud tops and cloud base heights, respectively. In this study (both Sc and Cu but except for the VOCALS TO flight), a single LCL is used for the cloud-base height for a given cloud-base level flight. However, the cloud tops have a 1-second resolution (from cloud radar). Thus, the cloud thickness has a 1-second resolution for the cloud-base level leg flights.

However, note that in case that the cloud radar is not operational such as VOCALS TO flights, both cloud tops and bases are estimated from the vertical structure of LWC, which has one value per cloud-base level-leg flight for a given day (daily resolution). Consequently, the cloud thickness is assumed to be constant during the cloud-base level flights, which are not as good as high resolution, and why we need either the cloud radar or G-band radiometer to measure cloud thickness and LWP directly at high resolution.

We examined the $S_o$ that calculated with the 1-second resolution of H, $N_d$, and $R$ by using cloud radar (both tops and bases where cloud bases have used the heights of cloud-base level leg flights). Although it is not shown in the paper, the results were robust. An example of $S_o$ that is calculated with a 1second resolution of cloud data for BACEX is shown here.

[Figure]

**Figure**. Precipitation susceptibility as calculated from BACEX aircraft data.

Note that the cloud thickness in the above figure is not the same as $H$ in the manuscript (e.g., Fig. 4) because the cloud bases in this figure have used the heights of cloud-base level leg flights, which are relatively constant during the flight is slightly higher than the actual cloud bases, resulting in overall lower H values.

P4 L28-29: Is using 1 second data appropriate? Given the sampling volume rate and scarcity of drizzle drops I wonder how statistically robust the R retrievals are. Just based on counting statistics, what are the measurement uncertainties in R? What is the theoretical minimum threshold on R given the sampling rate of 1 second?

Response: The CAS probe collects drops at 10 Hz then the DSDs are averaged to 1 Hz at each channel, whereas CIP collects drops at 1 Hz.

In the CIP and CAS (all the optical particle counters and size spectrometers), the counting statistics would be estimated for each channel by Poisons statistics, i,e. the square root of the total count. Thus for hundred particles in a channel, counting statistics would set the error at ± 10 counts, or 10 %. For 10 counts in the channel, the Poisson error is ±3.1 or 31%. If 1-second data is giving 10 particles in a channel, then the counting error is 31%, but by counting for ten seconds, you have reduced the error to 10%. In a channel where the count is 1 particle over the counting period, the error is 100%.

In a CN counter we don't try to determine size but count everything. There we have no issue because there are so many small particles that counting statistics is always satisfied. When we start analyzing the signals each particle generates, and trying to tell its size from the value of the signal we receive; then we run into counting statistics problems. We indicate that a particle that generates signals within a certain range belong to a size range and the narrower you make that range, the fewer particles you will sample in a given period of time. So if in any channel you have 100 particles, there would be 10% uncertainty in that number on account of Poisson statistics alone, then there will be an additional uncertainty due to viewing volume and electronic issues.

The manuscript is revised as "CIP acquire data every 1 second, but CAS probes acquire data every 10 Hz then the DSDs at each channel are averaged to 1 Hz. The cloud radar receives data every 3 Hz then is averaged to 1 Hz to pair with probe data."

P5 L3-6: Same question can be applied to z.

Response: Please see the above. In the revised manuscript we removed the Z-associated figure and text.

P6 L6: What are the 95 or 99% confidence intervals on this estimate? The scatter in Fig. 2 appears rather large.

Response: The confidence level is calculated from log ($N_d$) and −log(R) diagram such as Fig. 3.

P7 L2: A measure of the uncertainty will be helpful here as well.

Response: We added $r$ for the $S_o$ in the last paragraph on Page 6.

P7 L8: How were the H intervals chosen? Do the results vary with larger or smaller H bins?

Response: We chose the H intervals that include a similar number of data points in the H interval, and (at the same time) the H that gives the robust results even though we choose slightly different H. The effect of H interval on the $S_o$ estimates and it is discussed in Appendix C of the current manuscript

**Appendix C. The effect of H intervals on $S_o$ estimates.**

$S_o$ calculated with different H intervals can be seen by comparing Fig. 4 and Fig. A1 as an example. H intervals in Fig. 4(b) are about 30 m, while H intervals in Fig. A1 are about 50 m. The qualitative H-dependent behavior of $S_o$ is robust regardless of the chosen H intervals in case 1-second data are used. However, the chosen H interval may have effect on the estimate of $S_o$ that is calculated with a fewer data points, such as $S_o$ that is calculated with data averaged over the e-folding time.

The effect of H-intervals on $S_o$ estimates, which is estimated with data averaged over the e-folding time, is shown in Fig. B1. In summary, the results are robust regardless of H interval in general. However, if the H interval is chosen across the cloud thickness where the $S_o$ changes substantially (such as in which the cloud properties change substantially), the pattern of $S_o$ can be changed, indicating that the finer H interval would provide more accurate $S_o$. This is shown in Figs. 7 and 8. In Fig. 7, an H interval of 50 m hides the variation of $S_o$ between H 150 m and 200 m. The ln $(N_d)$ and −ln$(R)$ diagrams for H widths of 40 and 50 m are shown in Fig. 7. However, in case that the $S_o$ does not change substantially across the H intervals, the $S_o$ does not change even if the larger H interval is used (e.g., Fig. 8d). For example, $S_o$ calculated with subsets of data (e.g., 220 ≤H<250m, 250≤H<280m, 280≤H<310m) are about ~ 0.24 to 0.25. If the $S_o$ is estimated with all the data that fall into the three intervals (e.g., H > 200 m), the value is about 0.28, which is similar to three individual $S_o$ values. The results may indicate that the cloud properties such as cloud thickness where the cloud begins to precipitate could be of importance for accurate estimates of $S_o$ by affecting the optimal H interval and/or ranges.

[Figure]

**Figure C1**. $S_o$ is calculated with cloud data that are averaged over an e-folding time for E-PEACE. $S_o$ calculated with three H intervals (Δ30 m, Δ40 m, and Δ50 m) are shown. Horizontal bar indicates ±1σ cloud thickness for a given H interval.

[Figure]

**Figure C2**. The ln ($N_d$) and −ln ($R$) diagrams with fixed H intervals: (left) ΔH=40 m, (right) ΔH=50 m.

[Figure]

**Figure C3**. The ln ($N_d$) and −ln ($R$) diagrams with fixed H intervals (ΔH=30 m).

P7 L14-15: "We noted that S0 tended : : :. (not shown)." Why do the authors think that when the $N_d$ variations are small, S0 is high? Are N and h covariations leading to unrealistically high S0? Do uncertainties in the slopes increase in this case?

Response: We revised the sentence as follows: " We tested and applied a few criteria in the $S_o$ calculations, such as minimum $R$ thresholds, and the total number of cloud data points and spans of $N_d$ for a given $H$ interval. Based on these sensitivity tests, we calculated $S_o$ exclusively if $N_d$ varied a sufficient amount (e.g., dln($N_d$) spans at least 2.2) for a given $H$ interval since little variation of $N_d$ does not provide the proper perturbation of aerosols".

P7 L16: "if dlog(Nd) spans at least 2.2" Is the natural log used here? If so, what is the threshold of max(Nd)/min(Nd) used here? Perhaps around 10?

Response: It is natural log and yes it is around 9. We wanted to consider dataset only if $N_d$ varies large enough because $S_o$ metrics measure changes in $R$ with

changes in $N_d$. Thus, when $N_d$ changes little, it is not reasonable to calculate $S_o$. We removed the particular number in the revised manuscript. Further, we changed log to ln in the revised manuscript to avoid confusion.

P7 L16-17: "exceeds six for a given H" – this seems like a very small sample size for calculating slopes. Uncertainties in the slopes should be shown.

Response: There was hardly a case that had only 6 points. The criteria were used when we calculate $S_o$ in every 10 seconds, which corresponded to grey squares in the original Fig. 4. However, in a revised Fig. 4, we did not include gray squares because we additionally calculated $S_o$ over longer time-steps by considering the independence of the 1-second dataset. The smallest data points used for $S_o$ estimates (for Fig. 4) was 23 for BACEX in the lowest H interval and 10 for E-PEACE for the highest H interval. The sentence was removed from the revised manuscript as we no longer include the grey square symbols in Fig. 4.

P7 L20: "statistically significant at the 99% confidence level" - I suspect this means statistically significant with a comparison with a slope of 0. This is only the case if the each 1-second of data is independent of another. The authors need to demonstrate that this is the case, perhaps using the method of Leith (1973) or Bretherton (1999).

Response: We have addressed this by using longer time scales (e.g., e-folding time scale) and by resampling random flights and proved results are still robust.

Leith, C. E. (1973), The Standard Error of Time-Average Estimates of Climatic Means, J. Appl. Meteorl., 12, 1066–1069.
Bretherton, C. S. et al. (1999), The Effective Number of Spatial Degrees of Freedom of a Time-Varying Field, J. Clim., 12,7, 1990-2009.

P 7 L29-30: "$S_0$ tends to be overestimated: : :" Based on what has been shown so far, it doesn't appear that $S_0$ in necessarily overestimated when a larger 'averaging length scale' is used. It can be that $S_0$ is underestimated when every second of data are included.

Response: In Fig. 4, in fact, we did not average the length-scale. $S_o$ with the data of n=4 just indicates that we sub-sample the data every 4 seconds intervals (the data that chosen was still 1-second data but sub-sampled). Since the grey square symbols do not add any further insights to the Fig. 4, we removed the figure and associated text in the revised manuscript.

P7 L32: "H is estimated from the vertical structure of LWC for each day" - If daily mean H is used, then the sub-scale covariance between N and H should be examined, based on the other measurements. To what extent are H and N covarying and how can that potentially affect susceptibility estimates?

Response: We recalculated $S_o$ with data averaged over an e-folding time. By averaging over the cloud data, a cloud-mean H is used that is averaging over both H and N variations, so that their covariability is not consequential in this case (blue below). Further, the dependence of 1-second data is examined by comparing $S_o$ that calculated with 1-second dataset (grey in Fig. 6) with $S_o$ that calculated with data averaged over an e-folding time (blue in Fig. 6).

The overall pattern of the $S_o$ is robust (an increase and then a decrease) (Fig. 6). The main difference between two $S_o$ (by using 1-second data and by using data that averaged over the e-folding time) may be the leftward shift of H that peaks $S_o$. The e-folding time for VOCALS TO flights is summarized in Table 1.

[Figure]

**Figure 6**. $S_o$ for VOCALS TO flight is calculated with 1-second data (grey) and cloud data that are averaged over an e-folding time for each day (blue). The $\ln(N_d)$ and $-\ln(R)$ diagram is shown for each H interval. The horizontal bar in (a) indicates $\pm 1\sigma$. $S_o$ is calculated for the cloud data in groups with similar H (shown in Table 1).

P8 L1: "H of 9 Nov. (164+/- 18m)" To what extent is using daily mean H to group data appropriate? What is the true range of H from each day of flight? Are there cases where data from one day could potentially lie in a different bin?

Response: When the H is classified, we carefully chose the bins that are not overlapped withits neighboring bins; consequently, we only ended up with four groups. The cloud thickness for a given day is added in Table 1.

The daily mean H for VOCALS is included in Table 1.

**Table 1**. Dates used for this analysis during each experiment. Cloud thickness is shown (mean±1σ) for VOCALS with numbers of group category.

| No. | VOCALS (Sc) |
|-----|-------------|
| Period | Oct.-Nov., 2008 |
| Location | Southeast Pacific Sc decks |
| RF1 | 10/16 (2), 232 |
| RF2 | 10/18 (3), 292±22 |
| RF3 | 10/19 (3)323±16 |
| RF4 | 10/21 (1) 172 |
| RF5 | 10/22 (2) 224 |
| RF6 | 10/26 (2) 208 |
| RF7 | 10/27 (1) 142±38 |
| RF8 | 10/30 (2) 213 |
| RF9 | 11/1 (4) 641 |
| RF10 | 11/9 (1) 164±18 |
| RF11 | 11/10 (1) 194±21 |
| RF12 | 11/12 (2) 249 |
| RF13 | 11/13 (1) 183 |

P8 L3: "as the each" – remove "the"

Response: Removed "the" in the manuscript.

P9 L9: "no data were available between 800 m and 1500 m that satisfied the data analysis" – were there not enough data points that existed in this range to calculate the regressions or was the range of N too small? Could bins have been combined to get an estimate? In Fig. 5, it is difficult to make out much of a trend based on three susceptibility estimates.

Response: There was no data point (Please see Fig. 2d)

P9 L17-19: "The negative values of S0 in the largest: : : in that category" – based on the open circle designation (Fig. 4b), it appears that the susceptibility is statistically indistinguishable from zero, so there is no need to explain why it has a negative value.

Response: We reduced text by removing the details.

P9 L20-22: ": : : if the H varied substantially during the cloud-base level-leg flight on the day which S0 was calculated with a daily mean H." I wonder how the susceptibility estimates from the thinner clouds are similarly affected from the VOCALS flights. Can the authors point to any other data or references, which show that the cloud thickness variability in a flight day are smaller than the variability from flight to flight?

Response: This can be referred by Table 1 where the mean H (and 1σ) is shown. H variability within a day for the thinner cloud is larger than daily H variability, if $S_o$ is calculated with a daily mean H. For example, H variability on 10 Nov (RF11) is 21 m (and thus, H ranges 173-215m on 10 Nov.). H variability on 9 Nov (RF 10) is 18 m (and thus H ranges 146-182), indicating that the cloud thickness for these two days overlaps. However, it should be noted that the $S_o$ for VOCALS was calculated with a grouping method where we classified these two days into the same group (i.e., group1). Furthermore, cloud thickness in group1 and group2 for VOCALS does not overlap, and that's why we only had four H intervals for VOCALS dataset.
We agree with the reviewer that the thinner cloud may experience larger uncertainty of H than that in the thicker cloud. It may be possible that if one has a negative $S_o$ in thinner clouds, that could have resulted from the H uncertainty among many possible other sources of the error.

P10 L5-7: "probably show the impacts of meteorology on S0 within the fixed H, because the cloud data points close to each other with similar H are more likely to experience the same meteorology": Although the authors appear to argue that using larger averaging lengths lead to an overestimation of S0, can you not argue that S0 can be underestimated with a shorter averaging length due to covariance between N and H and smaller spatial scales?

Response: We removed $S_o$ calculated with n=1 to n=10 as it does not add more insight and is confusing.

P10 L7: The authors have addressed the similarity in the qualitative behavior of susceptibility between Cu and Sc clouds. Can they comment on how they agree in terms of absolute values? And at which thicknesses the peaks occur? Based on previous studies, is there a prior expectation of whether the peak should occur at the same location (H-value)?

Response: We cannot say anything about the absolute value of H where the $S_o$ peaks across the clouds. However, we suspect that the thickness at which the peaks occur could be related with the cloud thickness where the cloud precipitates. That being said we guess that $S_o$ peaks at a H value slightly higher than H where the cloud begins to precipitate. In fact, we are interested in examining this by using more observational datasets, by considering the normalized cloud depth since the cloud thickness varies spatiotemporally.

P11 L3-5: ": : : indicating a longer tc for the clouds sampled at mid-cloud level compared with those sampled at cloud-base." - In the developing stages of precipitation, this may be true, but if the drops start to fall out, they will eventually fall through bottom of the cloud, which means they will have a longer tc at the bottom of the cloud. One would expect the cloud base measurements to be a combination of parcels with short tc and with long tc.

Response: It is true that the $t_c$ (at cloud base and/or mid-cloud) is related to the cycle of clouds (cloud lifetime cycle) that were sampled. We removed $t_c$ and $Z$ related text from section 3.2 and revised the manuscript accordingly.

P11 L19-20: Include comparison and references to Mann et al., 2014 and Hill et al., 2015.

Response: The references are included.

P11 L27: "Note that not all of the data shown in Fig. 1 in Terai et al. (2012) are used for the S0 calculations in their study." Because their SR included a component coming from Sf, which took into account non-precipitating clouds, all of the data was used to calculate SR. Not all of the data was used for SI.

Response: We removed the sentence in the revised manuscript.

P12 L1-3: "As an example, this R threshold rejects all the data in Fig. 2a.." How about the H in the two datasets? Are there overlaps? Do the H and R values agree between the two studies?

Response: H overlaps between two of the datasets. However, the dataset from VOCALS (C-130) sampled cloud fields mainly in the open ocean (east to west direction) where POC dominates the clouds, and the clouds are thicker and precipitating (more intense and frequent precipitation). In contrast, VOCALS TO flights sampled the cloud fields near the continent where the environment was more polluted, less precipitating and consisted of smaller droplets compared with

clouds sampled from VOCALS C-130 flights. Therefore, even though the cloud overlaps in H and R somehow, clouds in TO flights are mainly thinner with less precipitation.

[Figure]

**Figure**. Cloud data for VOCALS C-130 flights (left) and VOCALS TO flights (right).

P12 L15-16: This is not the case. Their reflectivities were mainly taken from cloudbase retrievals and their N was the accumulation aerosol concentration, not the cloud droplet number.

Response: The arguments on the cloud-base and mid-cloud were removed in the revised manuscript as $R$ measurements (converted from $Z$) were taken from the cloud base.

P12 L25: Would the authors summarize the main conclusion of the paragraph? Is it that Terai et al. (2012) examined relationships in mid-cloud level, where accretion rates are high, and therefore, examine only the downward tail of susceptibility?

Response: We simply pointed out that the dataset from VOCALS C-130 flights sampled cloud field where accretion rates are high, and therefore, captures the downward tail of susceptibility predominantly. The argument of mid-cloud level versus cloud-base level associated with $t_c$ was removed and the manuscript is revised.

P13 L4-5: "This procedure can overestimate precipitation for a given Nd" – please elaborate on why this is the case. In many heavily precipitating clouds, the reflectivity is highest at cloud base.

Response: It is true that the reflectivity is highest at cloud base on many occasions in the heavily precipitating clouds, but not all the time. The actual $Z$ can be equal to

or less than the column maximum *Z* because the column maximum Z is the maximum Z along the column.

P13 L28-29: Although the use of sub-cloud aerosol concentration to calculate the susceptibility in Terai et al. (2012) might explain some of the differences between the susceptibility in this study and the susceptibility in their study, they did not make the assumption that the sub-cloud aerosol concentrations and cloud-base Nd were linearly related (see their Fig. 2 and the corresponding discussion).

Response: The manuscript is revised by focusing on some of the discrepancies between $S_o$ in current and Terai et al. could be contributed by the use of sub-cloud aerosols.

P14 L4: replace "Terei" with "Terai"

Response: Corrected.

P14 L6-7: "This method possibly could affect/change the slope: : :" One way to test this is to apply their method to the data in this study to determine whether susceptibility values using the data in this study are indeed overestimated when using their method.

Response: We removed the sentence as Terai et al. (2015) stated that the $S_o$ calculated from the linear regression of the bin mean $N_d$ and R shows nearly identical $S_o$ to that calculated from the tercile log difference method of Terai et al. (2012).

P14 L24-25: "We also note that Z increases with height that is consistent with the H dependent: : :" The Z will also increase with height just from an increase in condensed water that you would get from an adiabatic increase with height.

Response: We removed arguments on *Z* with heights in the revised manuscript.

P15 L26: "the lower R minimum threshold is desirable to use" – what threshold should the minimum be? Should sedimentation of cloud droplets be included? The appendix in Hill et al. 2015 suggests that the asymptotic limit of S0 for small LWP, where 'precipitation' is dominated by cloud sedimentation, is 2/3.

Response: We think the cloud droplets should include since susceptibility metrics explains of the second indirect effect.

Table 1: In the captions please specify what the numbers in parentheses mean.

Response: Revised as suggested.

Figure 2: What does the dashed line indicate?

Response: The dashed line indicates the Rainfall rate of 0.14 mm day$^{-1}$ and the changes are made in the manuscript.

Figure 3: Please provide the uncertainties in the slopes.

Response: The figure is revised.

Figure 4: What do the lighter pink colors indicate? As in my previous comment, the uncertainties in the slopes should be shown.

Response: Light pink indicates the $S_o$ for the KWACEX. The figure is modified in the revised manuscript.

Figure 5a: What do the horizontal bars indicate?

Response: One standard deviation of $D_e$. The figure caption is modified.

Figure 6: What do the horizontal bars indicate?

Response: One standard deviation of $D_e$. The figure caption is modified.

Figure 7a: What do A and B indicate?

Response: The figure caption is modified.
The authors prepare an analysis of precipitation susceptibility based on in situ measurements from four field campaigns with nearly identical aircraft payloads, two that sampled cumulus clouds and two that sampled stratocumulus. The authors report robust patterns, and advance hypotheses for the trends that they find, as well as possible explanations for differences between their findings and the results of past studies. The methodology for how to best make such calculations is not clearly settled upon, as evidenced by some of the sensitivity tests presented, but details are sufficient for this work to be reproduced. I rate that revisions to address the following comments can make this into a methodologically sound contribution that is suitable for publication. In particular, I think it needs to be taken into account whether 1-s samples are statistically independent when evaluating sample size. In addition, the authors seem to conflate sample size and horizontal scale dependence, which appears illogical. I also could not follow the arguments about

autoconversion versus accretion using this data set, leaving final statements seeming unsupported by the evidence provided. Comments are enumerated below.

1. There should be some guidance on the spatial scale over which equation 1 applies. A reader must assume implicitly based on this work that it is intended to apply to one second of flight time (100 m in horizontal and full vertical column)? Also a GCM grid cell (100 km in horizontal and full vertical column)? Really both identically? Please offer at least brief guidance for the reader in the introduction.

Response: revised as suggested in the introduction (near Eq. (1) and near the end of the introduction)

2. Can you pls comment on whether treating $N_d$ as a proxy for $N_a$ has any relevant consequences? For instance, does that proxy give stronger $S_o$ than using $N_a$ owing to a decreasing fraction of aerosol activated with $N_a$ increasing, all else being equal?

Response: We added the follows in the last section of the discussion: "In cases where sub-cloud aerosols are used for the $S_o$ estimates, these estimates give a smaller $S_o$ than those using $N_d$ due to the decreasing fraction of aerosol activated with $N_a$ increasing, all else being equal (e.g., Lu et al., 2009)"

3. I recommend revising the text to reflect the fact that not all current climate models use equation 2, such as those with prognostic precipitation species.

Response: Revised as suggested (near Eq. 2)

4. Should GCCM be GCM throughout?

Response: Corrected in the manuscript.

5. Using 1-second data, there is a big enough sample volume to accurately calculate Z from dZ/dD? For instance, can you show evidence that your 1-s sample volume is large enough to produce a smoothly continuous DSD? If everyone except me knows that this is possible, perhaps you can just point to a reference or provide a figure outside of the manuscript.

Response: In a revised manuscript we removed Fig. 5(b and c).

6. Page 6, line 17: It is stated that figure 2 "essentially shows that as Nd increases, R decreases." I would not jump to that conclusion from that figure. The amount of scatter around the trend in figure 3 demonstrates why. I would recommend leaving this statement out of the introduction to figure 2 and focusing instead on the fact that it shows well the range of R and Nd sampled during each experiment.

Response: Removed as suggested.

7. Page 7, paragraph containing line 25: Can the authors present evidence to indicate that sequential 1-s samples are statistically independent? It seems to me that a methodologically appropriate test of sample size for this study would be to randomly resample the data (if consecutive 1-s points are statistically independent) or else randomly resample the flights used (if they are not).

Response: We calculated $S_o$ by randomly resampling the flights as the reviewer suggested and the behavior of $S_o$ is robust (e.g., Fig. 8). The details are summarized in section 3.2 in the revised manuscript.

In this section, we estimated $S_o$ by randomly resampling the flights of E-PEACE to see whether the sequential 1-second samples are statistically independent. We mainly used 12 flights in this part (Fig. 6 and Fig. 8) to avoid further complication by including RF13 of which e-folding time of $N_d$ is much shorter than other flights (several seconds compared with 4-10 minutes of other flights. See Table 1 for the e-folding time).

$S_o$ calculated with random flights, at first glance, showed two distinctive types of behavior (Fig.7A). One is a similar pattern to that of the current $S_o$ (red and magenta in Fig. 7A) shown in Fig. 4 while the other is an almost constant $S_o$ near zero (blue and cyan in Fig. 7A).

[Figure]

**Figure 7A**. $S_o$ calculated from the randomly resample the flights.

The cloud data sampled during E-PEACE formed two groups (denoted as A and B in Figure 7). For example, group B lie in the lower- and left- side of the diagram that has lower R for the given $N_d$, but also includes top two highest R flights (RF 13 and RF3). Group B include RFs 3, 5, 6, 9, 10 and 13, and one whereas group A include flights 2, 4, 6, 7 11, and 12.

[Figure]

**Figure 7**. Daily mean values of $N_d$ and R for the 13 E-PEACE flights. Numbers indicate the flight numbers shown in Table 1.

The $S_o$ calculated with cloud data of group A and B is shown in Fig. 8. The $S_o$ pattern calculated with cloud data of group A is similar to $S_o$ shown in Fig. 4: $S_o$ is constant at lower $H$, followed by an increase then a decrease (Fig. 8a). In contrast, $S_o$ values calculated from group B were relatively constant near zero $S_o$ with the descending branch only (blue in Fig. 8c). It is of interest in Fig. 8 that if the $S_o$ is calculated with cloud data within the same category (upper or lower groups), the So shows the robust pattern (Fig. 8b and 8c). We did not examine here why the cloud properties in group A and B are substantially different, but it would be of interest for future work.

[Figure]

**Figure 8**. $S_o$ with cloud thickness for (a) 12 E-PEACE flights, for groups A and B shown in Fig. 7. (b) $S_o$ calculated with randomly resampled RFs within the group (b) A and (b) B.

Further analysis shows that the two RFs (RF13 and RF03) that have relatively small $N_d$ with high $R$ makes the differences in the $S_o$ pattern, depending on whether we include the data from those two low-$N_d$ with high $R$ into the dataset or not. For example, 6 odd flights (RFs, 1,3,5,7,9,11) and the first 6 RFs (RFs, 1-6) in Fig. 7A include RF03. Figure 9 also shows that if the $S_o$ is calculated with cloud data that do not include data from clean with heavy precipitating environments (i.e., RF13 and RF03), $S_o$ shows a similar pattern as that in Fig. 4. This also links to #11 (below) that shows how RF03 and RF13 changes the pattern of $S_o$.

[Figure]

**Figure 9**. The effects of RF03 and RF13 on $S_o$ estimates. (a) $S_o$ calculated for 13 flights during E-PEACE in addition to when either, or both, RF03 and RF13 are excluded. RF03 and RF 13 are the flights with high precipitation rates. (b) $S_o$ is calculated from group A with and without RF03 and RF13. $R$ and $N_d$ information for each flight is shown in Fig. 7.

8. Page 8, lines 9-11: Are the authors suggesting to use one LWP profile for each date for Sc and Cu? Unclear if this statement is limited to Sc.

Response:  The sentence applies to both Sc and Cu. However, no one wants to do this for Cu. The text was revised as follows: Nevertheless, if we calculate LWP by integrating LWC with height (e.g., in Sc), we would obtain one LWP profile that could be used for the entire cloud layer on a given day.

9. Page 10, first paragraph: This logic is not sound as currently written. First the authors state that So decreases with increasing sample size. There must be a limit to that if the system is well-defined and the significance of the results robustly evaluated, right? Then the authors compare such behavior to that found by others when decreasing the averaging length scale, which is a different issue entirely (see comment 1).

Response: The reviewer raises a good point and our original text was distracting. To address this point, we removed the time-averaging argument as it does not add any further insight in this study, but causes confusion.

10. 2nd paragraph of section 3.2: I really couldn't follow this paragraph. I would remove section 3.2 and figures 5 and 6 if the point of this paragraph can't be significantly clarified. Stating "But it is not discussed here." furthered this reader's impression that the Z analysis did not really add anything to this study.

Response: We removed arguments on the mid-cloud level versus cloud-base and the relevant argument as the process is associated with the cycle of cloud lifetime.

11. Page 11, line 23: There is no reason to show a figure such as A2. Simply state that results are insensitive. I would be much more interested to see a clear demonstration of a case where the R threshold is very important. It seems clear to the authors, but is not so clear to me how figure 4 would be affected, for instance.

Response: We replaced Fig. A2 with revised Fig. A2 that includes the $S_o$ calculated with different $R$ thresholds (R > 0.1, 0.5, 1.0, 1.5 and 2.5 mm/day). Figure A2 shows that $S_o$ become closer to zero as the R threshold increases (as So is not sensitive under the heavy precipitation). The Text has revised accordingly.

[Figure]

**Figure A2**. Precipitation susceptibility with the R thresholds.

Further, $S_o$ calculated with cloud data sampled during (1) RF13, (2) RF3, (3) RF13 and RF3, and (4) RFs 2,3,4, and 13 show the same results, which supports the insensitivity of $S_o$ under the high precipitation conditions (due to the dominance of accretion process) (Figure 11A shown below).

[Figure]

**Figure 11A**. So calculated with cloud data sampled during (a) RF 13, (b) RF 3, (c) RFs 3 and 13, and (d) RFs 2, 3, 4, and 13. The flights are chosen based on $N_d$ and R diagram where those flights sampled under the high precipitation conditions (The figure is not shown in the revised manuscript).

In addition, the effect of precipitation on the $S_o$ estimates is also shown in Fig. 9 (above #7).

12. Page 14, line 24: I really did not take away the autoconversion versus accretion behavior. There seemed to be a lot of handwaving in section 3.2. I basically feel that this statement is just not supported by the material shown. I think this needs to be much clarified or else removed.

Response: We removed $t_c$ and $Z$ related text from section 3.2 and revised the manuscript accordingly.

13. I don't understand the last sentence of the paper. The authors call for more studies on which range of H is most susceptible to preciptiation rate? This is a study on susceptibility of precipitation rate to Nd. Are the authors suggesting another thing? If the authors meant to further study So as defined here, why are further studies needed? All previous sentences in the last paragraph would indicate that the authors have already shown within which range of H Sc and Cu are most susceptible. Are these results somehow uncertain or incomplete? If that could be clarified and its relevance to the conclusions made here (regarding the general behavior of So in Sc and Cu found; is that uncertain?), I think that would better support this closing argument, if I understand it correctly.

Response: We removed the last sentence.

14. So many grammatical errors appear here in a paper with so many capable coauthors that I will not take my time to enumerate them, but merely note that this sometimes impacted my ability to evaluate the work (as in comment 13 above).

We carefully went through the text and corrected the language.

15. Please label $N_d$ axis units on figures 2 and 3.

Response: Labeled as suggested.

[revised manuscript text omitted]

---

## Author Response (AR2)

**Report #1**
Anonymous Referee #1

The authors have generally addressed my major concerns with the paper. In particular, the authors have addressed the issue of data independence and demonstrated that their general conclusions remain unchanged. The authors have also shown that H and $N_d$ are not correlated at the smaller spatial scales and are unlikely to affect the susceptibilities. There are still a couple minor instances where explanations are unclear or need some elaboration. Therefore, I recommend publication after these minor and technical comments have been addressed.

Minor comment

1) I do not quite understand the physical underpinning behind why the susceptibility should behave as they do, and I would like the authors to briefly address this. The three regimes of different process rates are used to explain the behavior of the susceptibility, and it is argued that at low LWP, not enough water is available to initiate rain. Indeed, the authors show that at low H, the susceptibility in many of the cases is indistinguishable from zero. Based on the measurements presented in this study, however, clouds with small H do appear to precipitate slightly ($> 0.01$ mm d$^{-1}$), which suggests that autoconversion process is active in clouds with small H. Should we not then expect these clouds to have a susceptibility greater than zero? Are clouds with R~0.01 mm d-1 not considered to be precipitating? Or are other factors, which appear as noise in the relationship, making it difficult to discern a relationship?

In Sc (Fig. 4b), $S_o$ is constant at about 0.2 for small H (e.g., H < ~230 m), whereas $S_o$ is close to zero (indistinguishable from zero) in Cu (Fig. 4a) for small H (e.g., H < ~500-600 m).
We did not study the characteristics of Sc clouds in detail, however, for Cu (e.g., clouds from BACEX in Fig. 4a), clouds begin to precipitate when H reaches 500-600m. Given that, Cu clouds of which H < 500-600m are (predominantly) non-precipitating clouds in general sense although still small portion of the clouds would possibly precipitate
The reason that the So in many of the cases is indistinguishable from zero is possibly due to the low threshold of R that we used to include weakly/barely precipitating clouds (R > 0.0001 mm/day), which is a quite low.

Technical issues

P3 L31-32: "the interrelationships examined are representative of GCM spatial resolution" – it is not clear what is meant here. Yes, the area covered by the field campaign are on the order of a GCM grid box, but the susceptibilities are calculated based on variations at much smaller spatial scales, which the GCM will not be able to resolve.

The manuscript is revised as "the mean interrelationships examined are representative of GCM spatial resolution". Although the individual points are representative of small spatial scales, any curve fits to these points or averaged values are representative of larger scales and the variability relative to the fit or the average represents the smaller scale variability.

P10 L25: Missing period between "accretion)" and "Here,"
Added in the manuscript.

P10 L29: Suggest inserting comma between "S0)" and "the accretion"
Added in the manuscript.

Table A1: It is my understanding that the correlations and P-values are calculated on the 1-sec data. Is this the case? If so, P-values are only meaningful when the data are all independent. So the correlations and the P-values should be calculated on the averaged data, there were averaged based on the e-folding timescale.

We included the P-value and correlation using 1-second data since $S_o$ with 1-second data and e-folding time remains unchanged. The number of data points for the averaged data is small (in particular the number of data points for a given H interval is considered), and thus, the correlation is hardly significant statistically.

Figure B2: Based on how little the susceptibility changed from no-threshold to R > 0.1 mm d$^{-1}$ threshold. Can't one infer that the threshold is not likely to be the reason behind the difference between the VOCALS data in this study and in Terai et al. (2012)?

Whether or not the R threshold alters the $S_o$ behavior more likely depend on the dataset (that are sampled). In the case of E-PEACE (Fig. 2c), the majority of cloud data has rainrate larger than 0.1 mm/day. On the other hand, in the case of VOCALS-TO flights (Fig. 2a), a lot of cloud data have less than 0.1 mm/day.

[Figure]

**Figure 2:** Scatter diagrams of cloud droplet number concentrations $N_d$ and precipitation, $R$, for E-PEACE (left) and VOCLAS-TO Flights (right). Colors indicate cloud thickness $H$. The dashed line indicates an $R$ value of 0.14 mm day$^{-1}$.

In the case of Terai et al. (2012), (Please see figure below), $S_I$ is calculated with data R > 0.14 mm/day that excluded all navy-blue-ish data in Fig. 1 from Terai et al. (2012). If Terai et al. (2012) included all the dataset of R > 0.001 mm/day, their $S_o$ pattern would have included as follows. i) $S_o$ is insensitive to H,

for H < 200 m (as aerosol increases R is about constant-all navy colors), and ii) $S_o$ increases with H for 200 < H < 300 m (as aerosol increases R decreases).

[Figure]

Fig.1 from Terai et al. (2012)

Figure 11: In the text, it is incorrectly referenced as Fig. 7a. And in Fig. 11b and 11c, wouldn't scavenging of aerosols lead to a change in Nd, rather than in R as the arrows indicate?

Corrected in the manuscript.
The arrows in Fig. 11 indicate the changes in $S_o$. The effect of wet scavenging is shown in Fig. 11a (Changes in $N_d$ from B to A for a given R, which artificially can provide higher $S_o$)

**Report #2**
Anonymous Referee #3
**Suggestions for revision or reasons for rejection**
I think the authors have done a nice job on addressing comments from two reviewers. Here I have some further comments for authors to consider:

**Page 2, Lines 7-8**: The difference of Si and S0 noted here is confusing and may not be correct. My understanding S0 is calculated for clouds that are precipitating. If a cloud does not precipitate, R is 0 and you can not calculate S0 (remember it is ln(R) is used in Eq. (1)). So it is confusing to see the statement of "SI is equivalent to S0 only when 100% of the sampled clouds are precipitating". The only difference here is how large a threshold of rainrate is applied for calculating S0 or SI (They are completely equivalent).
The manuscript is revised accordingly.

Page 2, Lines 16-17, and Eq .(2): "When GCMs consider aerosols, the rainrate R is often parameterized in terms of LWP and Nd as Eq. (2). …". This statement is inaccurate. Rainrate from liquid clouds are usually from two terms, one is autoconverion and the other is from accretion. The autoconversion term is typically parameterized using Eq. (2), but not for accretion. So this needs to be clarified here.

Manuscript is revised as follows: In global climate models (GCMs), aerosol effects on rainrate are represented by either a prognostic scheme or an empirical diagnostic scheme. When GCMs consider aerosols, the rainrate $R$ is often parameterized in terms of LWP and $N_d$ as Eq. (2)

$$R = LWP^{\alpha} N_d^{-\beta} . \tag{2}$$

Climate models typically assume a fixed value of the autoconversion parameter ($\beta$ in Eq. 2), ranging between approximately 0 and 2 (e.g., Rasch and Kristjansson, 1998; Khairoutdinov and Kogan, 2000; Jones et al., 2001; Rotstayn and Liu, 2005; Takemura et al., 2005). "Readers should note that rainrate from liquid clouds are usually from two terms; one is from autoconversion, and the other is from accretion (see Sect. 3.3). Since $S_o$ in Eq. (1) includes contributions from both autoconversion and accretion, in the case where accretion has little contribution to rainrate, $S_o$ may then be equivalent to the exponent $\beta$ in Eq. (2) at fixed LWP". Field studies of precipitating stratocumulus (Sc) clouds have reported $\beta$ values ranging from 0.8 to 1.75 at fixed LWP (e.g., Pawlowska and Brenguier, 2003; Comstock et al., 2004; vanZanten and Stevens, 2005; Lu et al., 2009). Such single power law fits, however, do not capture the changes in $S_o$ with LWP or $H$, which is important since previous works have revealed that the response of cloud rainrates to aerosol perturbations vary as a function of LWP (or $H$).

Page 2, Line 21: "S0 in Eq. (1) is equivalent to the exponent beta in eq. (2) at fixed LWP". This statement is incorrect. As I note in my comments above, rainrate comes from two parts, one is from autoconverion and the other is from accretion. But beta in eq. (2) is for autoconversion. In S0, it includes contributions from both autoconverion and accretion. So only in the case where accretion has little contribution to rainrate, S0 may then be equal to beta.
Please see above.

Page 7, Line 22-23: Again, the statement 'The low R threshold is intended to include both non-precipitating and precipitating clouds' is confusing. I do not think non-precipitating clouds can be included by using a low R threshold. You will have to choose clouds with a certain R threshold. Even though this threshold can be very low, there are still clouds that are not included. Also, as a threshold of 0.001 mm/day is really low here, I wonder what is the uncertainty in your rainrate calculation. How this might affect the retrieval of rainrate that is low.

The manuscript is revised as "The low R threshold is chosen to include precipitating and very lightly participating clouds. The 0.001 mm/day threshold is indeed very low; the uncertainty in rainrate calculation is larger than 0.001 mm/day threshold. For all intents and purposes, the 0.001 mm/day threshold is equivalent to no precipitation.

Page 8, Line 16: "one LWP value for the entire cloud layer on a given day". This is not clear to me. Please clarify.
We removed the sentence to clarify it.

Page 9, e-folding time. The authors cited Leith (1973) for e-folding time. Please elaborate here what is the e-folding time in this context and how this is calculated, so readers may not have to refer back to Leith (1973) understanding this.
The manuscript is revised as "First, we calculated $S_o$ by considering the e-folding time scale (Leith, 1973) in which an autocorrelation decreases by a factor of e".

Page 12, line 16: Again, I think it is not appropriate to use "non-precipitating" here.
The manuscript is revised as "Therefore, to study the extent that aerosols suppress precipitation, it would be more appropriate to encompass the full range of weakly to heavily precipitating clouds that include both

autoconversion and accretion processes".

Page 14, line 7-8: the comparison between S0 and beta. As I noted above, S0 and beta are not the same thing, and you can not directly relate S0 with beta here. So the statement here is not supported.
We removed the sentence.

Two missing references: Hill et al. (2015) and Wang et al. (2012)
The references are added in the manuscript.

[revised manuscript text omitted]